# Hierarchical Retrieval at Scale: Bridging Interpretability and Efficiency

**Shubham Gupta** [1 2 3] **Zichao Li** [1 4 3] **Tianyi Chen** [1] **Cem Subakan** [2 3]
**Siva Reddy** [1 4 3] **Perouz Taslakian** [1 4 3] **Valentina Zantedeschi** [1 2]

## Abstract

Information retrieval is a core component of many intelligent systems as it enables conditioning of outputs on new and large-scale datasets. While effective, the standard practice of encoding data into high-dimensional representations for similarity search entails large memory and compute footprints, and also makes it hard to inspect the inner workings of the system. Hierarchical retrieval methods offer an interpretable alternative by organizing data at multiple granular levels, yet do not match the efficiency and performance of flat retrieval approaches. In this paper, we propose RETREEVER, a tree-based method that makes hierarchical retrieval viable at scale by directly optimizing its structure for retrieval performance while naturally providing interpretability through meaningful semantic groupings. Our method offers the flexibility to balance cost and utility by indexing data using representations from *any* tree level. We show that RETREEVER delivers strong coarse (intermediate levels) and fine representations (terminal level), while achieving the highest retrieval accuracy at the lowest latency among hierarchical methods. These results demonstrate that this family of techniques is viable in practical applications [1].

## 1. Introduction

Modern intelligent systems increasingly rely on retrieval mechanisms to condition their outputs on new and large-scale datasets. In particular, retrieval enables Large Language Models (LLMs) to generate content grounded in external knowledge, thereby alleviating hallucinations (Shus-

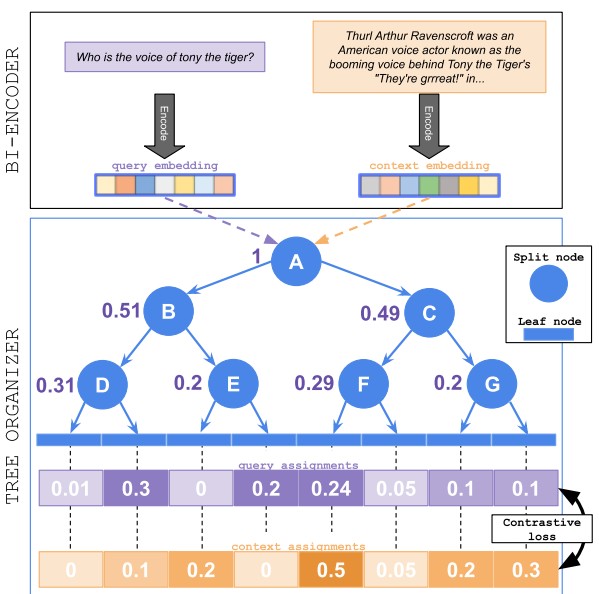

*Figure 1.* RETREEVER architecture and traversal: During training, a positive query–reference pair is encoded by the frozen Bi-Encoder $E$; these embeddings pass through each split node of a tree to yield left/right routing probabilities, which are merged into a leaf-assignment vector (one probability per leaf). At inference, leaf-level probabilities provide fine representations and intermediate-node probabilities serve as coarser representations.

ter et al., 2021), while scaling to massive datasets without increasing the number of internal parameters. As such, retrievers have become a key component in transforming LLMs into tailored experts without the need for exhaustive fine-tuning (Lewis et al., 2020; Izacard and Grave, 2021; Gao et al., 2023). A popular retrieval approach consists of representing items as high-dimensional embeddings and use nearest-neighbor techniques to find relevant content for a given query embedding (Karpukhin et al., 2020). While effective, the high dimensionality of these embeddings results in a large memory footprint and heavy computation at inference time. The items are also stored without any human-readable structure or understandable grouping, making it difficult to derive insights from the data or verify the retrieval process.

In this paper, we propose RETREEVER, a tree-based method where documents are organized and represented at various granular levels, offering coarse-to-fine representations,

[1]ServiceNow Research [2]Université Laval [3]Mila – Québec Artificial Intelligence Institute [4]McGill University. Correspondence to: Shubham Gupta <shgup1@ulaval.ca>.

*Proceedings of the 43rd International Conference on Machine Learning*, Seoul, South Korea. PMLR 306, 2026. Copyright 2026 by the author(s).

[1]Our code, models and demos are available at https://github.com/ServiceNow/retreever

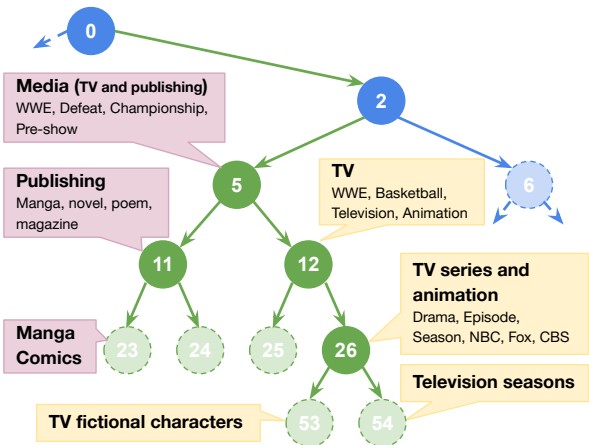

*Figure 2.* Inspecting a RETREEVER tree learned on NQ: Visualization of the topics (bold) and keywords extracted from the contexts assigned to one subtree (in green) rooted at node 5 of a RETREEVER tree of depth 10 trained on NQ. For compactness, we represent only a subset of the nodes, and stop at depth 5. Topics are locally coherent along a path, indicating that RETREEVER naturally groups contexts semantically.

learned entirely end-to-end by optimizing for retrieval performance as the learning objective. These representations offer the flexibility to balance cost and utility. Rather than training separate models for each size, a single full-capacity tree is learned, and at inference one can simply choose a tree level, higher for richer longer embeddings, and lower for smaller coarser ones, so content can be encoded at different sizes to match available computational budget. RETREEVER takes inspiration from the hierarchical retrieval literature, where tree organizers are built and navigated via calls to an LLM by cross-encoding textual queries and contexts (Chen et al., 2023; Sarthi et al., 2024; Edge et al., 2024). Such retrievers have the advantage of preserving the existing organization of the data, such as document divisions and entity relationships, and grouping documents semantically into a readable tree organization. However, their reliance on LLMs makes them slow and expensive to train and run, hence impractical even for medium-size text corpora. In contrast, RETREEVER is designed to operate entirely on embeddings, eliminating the need for LLM calls during both construction and navigation.

Figure 1 shows a schematic of our approach for document retrieval. RETREEVER first converts items (i.e., queries and references) into an embedding using a standard encoder, such as BERT (Devlin et al., 2019) for text and DINOv2 (Oquab et al., 2023) for images. It then uses these representations to learn a binary tree structure end-to-end by optimizing directly the retrieval objective, making use of the query-reference supervision available from most datasets. This is achieved by learning the parameters of a routing function at each internal node of the tree, such that positive query-reference pairs are routed similarly through it. The learned hierarchical structure naturally provides an organi-

zation of the items, which allows us to probe the tree to gain insights into the database and retrieval operations. Practitioners can inspect different levels of the tree to identify key thematic clusters influencing retrieval or understand why certain branches lead to specific content. While this is not a complete causal explanation of content matching, it does ease the inspection of the model's inner workings. In Section 5, we analyze nodes at various levels of the hierarchy and show that documents in these nodes have thematic overlaps even though the tree is solely optimized for query-document similarity without any clustering objective.

Our contributions are as follows:

1. We propose RETREEVER, a hierarchical and inspectable retriever that learns coarse-to-fine representations aligned with levels of a retrieval-optimized tree. Unlike existing hierarchical retrievers, RETREEVER scales to large corpora without requiring LLM calls.

2. We show that RETREEVER achieves the best latency and performance trade-off w.r.t both hierarchical tree-based and representation-based retrievers. Such results are consistent across text, image and speech datasets, demonstrating that our framework is modality-agnostic.

3. We highlight how RETREEVER's learned hierarchy induces semantically coherent groupings that can be directly inspected by humans, providing an interpretable window into the retrieval process and supporting better downstream decisions.

## 2. Related Work

**Bi-encoders** Retrieval models typically rely on sparse or dense vector embeddings to select the most relevant items for a query, reducing the problem to a nearest-neighbor search. Dense retrieval models (Karpukhin et al., 2020; Khattab and Zaharia, 2020) rely on encoders, neural networks designed to embed the semantic meaning of text into dense vector representations. State-of-the-art sentence embedding models often leverage bidirectional encoders, such as BERT (Devlin et al., 2019) and BGE (Xiao et al., 2024), while others are built on pretrained large language models, such as LLM2Vec (BehnamGhader et al., 2024). Some encoders, such as BGE-large (Xiao et al., 2024), are specifically designed for retrieval, offering a sentence embedding framework optimized for such tasks. Other notable examples are: DPR (Karpukhin et al., 2020), which finetunes a BERT dual encoder to separately encode queries and documents into a single dense vector by contrastive learning; ColBERT (Khattab and Zaharia, 2020), which finetunes BERT to encode queries and documents into multiple dense vector representations and by matching individual token embeddings between queries and documents. Unlike sparse retrieval methods, such as BM25 or TF-IDF (Salton and Buckley, 1988), these dense representations are less inter-

pretable, expensive to compute (generally because of attention operations) and take up a large amount of storage space. They do however perform better on many downstream tasks such as QA, specially for complex queries.

**Vision and audio encoders** Beyond text, dense representation learning has been widely studied for visual and auditory modalities. In vision, convolutional architectures such as ResNet-50 (He et al., 2016) have long served as strong baselines for image embeddings, while transformer-based models such as the Vision Transformer (ViT) (Dosovitskiy et al., 2021) have become a dominant architecture for learning global visual representations. Self-supervised approaches built on ViT, such as DINOv2 (Oquab et al., 2023), produce high-quality image embeddings that transfer well across a wide range of downstream tasks without task-specific supervision. Multimodal models such as CLIP (Radford et al., 2021a) further extend ViT-based encoders by jointly embedding images and text into a shared representation space, enabling cross-modal retrieval. For audio, transformer-based encoders have similarly emerged as the standard approach. Audio Spectrogram Transformers (AST) (Gong et al., 2021a) adapt the ViT architecture to time-frequency representations and achieve strong performance on large-scale benchmarks. As with text and vision encoders, these models provide compact representations suitable for similarity search, but incur significant costs and offer limited interpretability.

**Hierarchical retrieval** Hierarchical retrieval approaches are designed for text retrieval, and aim to balance test-time efficiency and accuracy by organizing the retrieval process into multiple stages, typically via coarse-to-fine filtering of candidate documents. Many hierarchical methods face challenges related to computational cost and performance trade-offs, and scale poorly with the size of the corpus. MemWalker (Chen et al., 2023) segments long texts into chunks, summarizes them into nodes that are recursively merged into a hierarchical memory tree, and at query time traverses this tree via iterative prompting to stay within the LLM's context window. Raptor (Sarthi et al., 2024) uses a clustering algorithm to group similar documents and then applies an LLM to recursively summarize and re-embed chunks of text, constructing a tree with differing levels of summarization from the bottom up, resulting in a structured, multi-layered tree representation of the original documents. GraphRAG (Edge et al., 2024) uses an LLM to build a graph-based text index by first deriving an entity knowledge graph from source documents and then pregenerating community summaries for related entities. For a given question, partial responses are generated from each community summary and combined into a final summarized response. While hierarchical retrieval methods improve response quality using LLMs and provides inspectable structures, they incur high computational costs and slow processing times, especially

with large datasets. This trade-off makes them less suitable for real-time or resource-limited scenarios, emphasizing the need for more efficient solutions. RETREEVER overcomes these limitations by constructing and navigating a tree with no LLM calls.

**Coarse-to-fine representations** As compute and running time generally scale with the representation size, an effective solution to limit retrieval costs is through dimensionality reduction(Van Der Maaten et al., 2009). When the computational budget is not known in advance, the standard solution is to train multiple models or low-dimensional adapters not to incur into accuracy degradation, as one would by applying post-hoc compression techniques. However, this solution requires higher training and memory costs than learning and storing a single model. Single-model alternatives have been recently developed (Yu et al.; Cai et al.; Kusupati et al., 2022). In particular, (Kusupati et al., 2022) propose Matryoshka Representation Learning (MRL), a simple framework that learns a nested representation. MRL boils down to learning an adaptive capacity embedding, ensuring that any first m-dimensions vector is as accurate as an independently trained m-dimensional one. This approach improves retrieval efficiency and flexibility, making it well-suited for diverse and evolving retrieval scenarios. Similarly, RETREEVER benefits from such advantages by training a nested representation, where each level input-to-node assignment corresponds to an embedding. Its structure and the learning of assignments additionally provide an organization of the documents into semantic groupings, allowing practitioners to inspect the corpus content and the inner workings of the retrieval system.

**Differentiable trees and hierarchical indexes** Because of their discrete form, tree structures have been traditionally optimized by greedy algorithms, specialized for a particular objective, e.g. for classification, regression, or hierarchical clustering(Quinlan, 1986; Krishnamurthy et al., 2012; Quinlan, 2014; Breiman, 2017; Moseley and Wang, 2023). Recent literature has focused on differentiable formulations to take advantage of continuous optimization techniques for training trees on large datasets and for arbitrary objectives(Irsoy et al., 2012; Yang et al., 2018a; Monath et al., 2019; Tanno et al., 2019; Zantedeschi et al., 2021; Marton et al., 2024). We leverage this literature and extend it to learning trees that are optimal for the retrieval objective (via contrastive learning), and with complex split and propagation functions. Finally, trees and graphs have been used in the retrieval literature as indices for storing and quickly retrieving document embeddings via approximate similarity search, and not designed for being inspected(Bernhardsson, 2017; Malkov and Yashunin, 2018; Douze et al., 2024). RETREEVER does not belong to this line of works, as it is an embedding extractor that learns an inspectable tree to organize and represent documents at different granularity

and into semantic groupings.

# 3. Proposed Method

RETREEVER consists of (1) a frozen encoder $E$ that produces embeddings for any text input, and (2) a learnable binary tree $T$ that serves as a shared router for both queries and references, organizing references into a hierarchy and routing queries to the relevant region of the tree (see Figure 1). In this section, we describe how the tree hierarchy is learned by continuous optimization and how search is performed at inference. In what follows, we use the term *context* to refer to the content organized by RETREEVER, which could be images, reference documents or audio recordings.

## 3.1. RETREEVER Architecture

**Tree Formulation.** Let $\{(\mathbf{q}_i, \mathbf{c}_i) \in \mathcal{P}\}_{i=1}^n$ be positive query-context pairs where $\mathbf{q}_i$, $\mathbf{c}_i$ are respectively the query and context embeddings generated by the encoder $E$. Our goal is to learn a binary tree $T$ that assigns $\mathbf{q}_i$ and $\mathbf{c}_i$ to the same leaf node, so that retrieval accuracy is maximized. Such leaf assignments, denoted $T(\mathbf{x}_i)$, are obtained by routing a given input $\mathbf{x}_i \in \mathcal{X}$ (e.g., $\mathbf{x}_i := \mathbf{q}_i$ the query embedding) sequentially through branching nodes until it reaches the terminal level of the tree. Specifically, we parametrize each branching node $t$ by a split function $s_{\theta_t}$ that routes an input to either its left or right child.

Hard context-to-node assignments would result in piecewise constant functions with null gradients, making back-propagation ineffective. Therefore, we make use of probabilistic relaxations, inspired by works such as that of Irsoy et al. (2012), to make the tree learning problem differentiable, hence allowing the learning of the split parameters jointly by gradient descent so that they are optimal for retrieval. In such a relaxation, a split function $s_{\theta_t}$ outputs a soft routing score rather than a hard left/right decision. This replaces hard routes through the tree with a probability distribution over the $|T|$ nodes of the tree, encoded by a vector $\mathbf{z}_i \in [0,1]^{|T|}$, where each entry $\mathbf{z}_{i,t}$ denotes the probability that input $i$ reaches node $t$ (see Figure 1).

**Choice of Split Function.** A split function $s_{\theta_t} : \mathcal{X} \to [0,1]$ can be implemented in several ways, as long as it outputs a routing score that determines the probability of routing an input left or right. Its simplest form is a linear projection, such as the one used in Zantedeschi et al. (2021): given a split node $t$, its left child $t_{\text{left}}$ and right child $t_{\text{right}}$, the split function is defined as the dot product $s_{\theta_t}(\mathbf{x}_i) = \theta_t \mathbf{x}^T$ (interpreting $\theta_t \in \mathcal{X}$ as a hyper-plane). No matter the design of split function, the routing probabilities are finally computed as $\mathbf{z}_{t_{\text{left}}} = \sigma(s_{\theta_t}(\mathbf{x}_i))$ and $\mathbf{z}_{t_{\text{right}}} = 1 - \sigma(s_{\theta_t}(\mathbf{x}_i))$, with $\sigma(x) := \frac{1}{1+e^{-x}}$ the sigmoid function, to ensure a valid probability distribution.

While the linear split function is computationally efficient, it may lack the expressiveness to capture complex routing decisions. We therefore also explore neural network-based split functions. In particular, we propose a **cross-attention-based function** (Vaswani, 2017) that operates directly on the token-level representations from the encoder, where learnable tree embeddings attend to the input sequence $\mathbf{x}_i \in \mathbb{R}^{n_d \times d}$ of $n_d$ token embeddings. The cross-attention split function (see Appendix A.1.3 for more details) is generally more expressive than a linear projection (as our ablation shows in Appendix A.1). Indeed, the tree embeddings and projection matrices act as memories that store information from past query and context embeddings, useful for scoring the current inputs.

**Tree Propagation** Note that any split function defined above outputs a score that is independent of the scores from the other nodes. However, the probability of reaching a node should be constrained by the probability of having reached its parent, the parent of its parent, and so on, to avoid degenerate trees where a descendant has higher probability of being reached than its ancestors. The simplest way of enforcing such tree constraints is by propagating the scores of the ancestors down to the node of interest by multiplying the probabilities along the path. We refer to this type of tree propagation as *product propagation* and use it as default method in our experiments. Given a node $t$ and its ancestors $\mathcal{A}_t$ (the nodes along the path from the root to $t$), the probability of input $x_i$ reaching $t$ is $T(\mathbf{x}_i)_t = \mathbf{z}_t \prod_{a \in \mathcal{A}_t} \mathbf{z}_a$. Notice that the product propagation naturally enforces that the sum of node probabilities at any depth is always constant and equal to 1. Alternatively, one can learn how to propagate probability scores through the tree. We refer to this type of tree propagation as *path consistency propagation* and describe it in Appendix J. To keep both training and inference efficient even for deep trees, we compute all split scores in parallel before applying the chosen propagation scheme.

## 3.2. Training by Contrastive Learning

The task now is to learn end-to-end the parameters $\Theta$ (that include those of the split and propagation functions, and of the tree embeddings), so that any query is routed optimally to the leaf that contains its corresponding ground-truth context. Such an optimal assignment is achieved when positive query-context pairs are independently routed similarly through the tree, leading to similar leaf assignments. However, optimizing solely for perfect positive assignments is likely to lead to the trivial solution where all contexts and queries are routed to a single leaf, resulting in a collapsed tree with a single active path. To avoid such a solution, we optimize a contrastive objective that implicitly encourages the model to use the tree non-trivially by spreading unre-

lated queries and contexts across different leaves. Building on contrastive learning literature (Oord et al., 2018; Radford et al., 2021b), we optimize the following Softmax-based InfoNCE loss,

$$-\frac{1}{2|\mathcal{P}|}\sum_{i=1}^{|\mathcal{P}|}\left(\log\frac{e^{\text{sim}(\mathbf{q}_i,\mathbf{c}_i)}}{\sum_{j=1}^{|\mathcal{P}|}e^{\text{sim}(\mathbf{q}_i,\mathbf{c}_j)}}+\log\frac{e^{\text{sim}(\mathbf{c}_i,\mathbf{q}_i)}}{\sum_{j=1}^{|\mathcal{P}|}e^{\text{sim}(\mathbf{c}_i,\mathbf{q}_j)}}\right) \tag{1}$$

where $\mathcal{P}$ is the set of query-context pairs, and we take the similarity $\text{sim} : [0,1]^{|\mathcal{T}_L|} \times [0,1]^{|\mathcal{T}_L|} \to [-1,0]$ to be the negative Total Variation Distance (nTVD) between two leaf assignments: $\text{sim}(\mathbf{a},\mathbf{b}) = -\frac{1}{2}\sum_{l=1}^{|\mathcal{T}_L|}|a_l - b_l|$, where $\mathbf{a} = (a_1, a_2, \ldots, a_{|\mathcal{T}_L|})$ and $\mathbf{b} = (b_1, b_2, \ldots, b_{|\mathcal{T}_L|})$ are the leaf assignment distributions (i.e., the restriction of $T(x_i)$) to the leaf nodes, ignoring internal nodes), and $|\mathcal{T}_L|$ is the number of leaf nodes in the tree. In Eq.(1), the left term encourages any query to have a leaf assignment similar to the assignment of its ground-truth context and different from any other context in the batch. Conversely, the second term encourages contexts to be routed similarly to their positive queries and differently from their negative ones. We do not introduce an explicit routing-temperature schedule or load-balancing regularizer; instead, degenerate collapse is discouraged implicitly by the bidirectional InfoNCE objective, which contrasts each positive pair against many in-batch negatives, together with a learnable softmax temperature.

**Stochastic Depth Training**   The contrastive loss in Eq.(1) is defined over leaf assignments. However, due to the hierarchical structure of the tree, optimizing at the leaf level implicitly shapes the assignments at intermediate levels as well, since each intermediate node's probability is the sum of its descendants' probabilities. To further strengthen these intermediate representations for inference (see Section 3.3), we introduce a training scheme where, at each iteration, a tree level is sampled uniformly at random and the contrastive loss is optimized with respect to assignments at that level. We refer to this as stochastic depth training, in contrast to constant depth training where optimization is performed exclusively at the leaf level. We denote models trained with the stochastic scheme as RETREEVER-STOCHASTIC.

### 3.3. Inference, Interpretability, and Latency

Once the tree is trained, we index new content by encoding each reference item with the encoder $E$ and then routing it through the tree $T$ to obtain node assignments.

**Coarse-to-Fine Representations**   Routing an encoded text through the tree yields hierarchical representations at multiple granularities. At each level, the assignment vector captures alignment with learned semantic groups, with dimensionality determined by the number of nodes at that level. The leaf-level representation has dimensionality $|\mathcal{T}_L|$,

providing the finest semantic granularity. Intermediate levels offer progressively coarser representations (e.g., dimension 16 at level 4), enabling a flexible trade-off between representation cost and retrieval accuracy.

**Indexing and Search**   For retrieval, we consider two indexing strategies. In *single-index search*, we build a vector index based on the chosen level assignments, process the query, and return its nearest neighbors based on the nTVD metric used at training. In *multi-index search*, we exploit the implicit clustering induced by the tree: each document is assigned to the leaf node for which it has the highest probability, and we build a separate index for each leaf cluster. At query time, we route the query through the tree, select its top-$k$ leaves, and search only the corresponding indices. This reduces the number of documents examined, trading off recall for latency. We report data access percentage (fraction of the corpus searched) as a proxy for latency in our experiments.

**Implicit Semantic Coherence**   Learning node assignment probabilities, as opposed to unconstrained features, naturally provides a hierarchical and soft clustering of the documents that can be inspected by the user. We show in Section 5 that RETREEVER's learned clusterings capture semantic groupings, where assigned documents share topics and keywords, even though RETREEVER does not directly optimize for semantic coherence.

## 4. Retrieval Experiments

In this section, we assess the retrieval performance of RE-TREEVER at different representation levels and data access rates, as compared to flat and hierarchical retrieval methods. Our results indicate that using the learned node assignments for retrieval (1) generally improves the representation power of the encoder at any representation level, (2) outperforms traditional approximate search even at low data access rates, (3) strikes the best latency-accuracy trade-off among hierarchical retrieval methods and (4) provides an inspectable and meaningful organization of content, making it much more transparent than similarity search based methods.

### 4.1. Metrics and datasets

To evaluate retrieval results, we measure the following standard metrics: RECALL@K, which assesses the proportion of ground-truth documents that are present within the top-k results returned by the retriever; *Normalized Discounted Cumulative Gain* at rank $k$ (NDCG@K) which accounts for ground-truth ranking, as relevant documents appearing earlier are considered more valuable. Our main benchmark includes four textual question-answering datasets ( NATU-RAL QUESTIONS (NQ) (Kwiatkowski et al., 2019), HOT-

*Table 1.* Average retrieval latency (ms) and NDCG@10 (%) for tree-based methods on text retrieval tasks. RETREEVER strikes the best latency-performance trade-off among hierarchical retrievers.

| METHOD | NQ | | HOTPOTQA | | TOPIOCQA | | REPLIQA | |
| --- | --- | --- | --- | --- | --- | --- | --- | --- |
| | LAT. ↓ | NDCG@10 ↑ | LAT. ↓ | NDCG@10 ↑ | LAT. ↓ | NDCG@10 ↑ | LAT. ↓ | NDCG@10 ↑ |
| HIER-KMEANS | 4.60 | 14.87 | 3.82 | 27.34 | 3.71 | 7.50 | 3.70 | 27.06 |
| HIER-GMM | 5.42 | 11.15 | 5.06 | 17.19 | 3.81 | 4.54 | 8.67 | 14.03 |
| RAPTOR | 1758.05 | 40.28 | 2246.51 | 59.37 | 701.61 | 18.69 | 1647.18 | 49.58 |
| RETREEVER | 10.65 | **54.48** | 10.01 | **85.04** | 9.66 | **31.00** | 9.47 | **80.23** |

POTQA (Yang et al., 2018b), TOPIOCQA (Adlakha et al., 2022), and REPLIQA (Monteiro et al., 2024b)), one image dataset (IMAGENET-1K (Deng et al., 2009)) and one speech dataset (VOXCELEB2 (Chung et al., 2018)). To further stress-test RETREEVER in a substantially larger retrieval setting, we also evaluate it on the REDDIT TITLE–BODY corpus,[2] (Baumgartner et al., 2020) a processed collection of Reddit submission titles paired with their post bodies; we report the full setup and results in Appendix I. For text retrieval datasets composed of natural-language query–reference pairs, we follow the pre-processing procedure of Monteiro et al. (2024a), except that for HOTPOTQA we concatenate all relevant contexts and discard irrelevant ones to obtain the reference. For the Reddit scaling study, we treat each submission title as the query and the corresponding post body as the reference context. For image and speech retrieval, we construct $m$ positive query-reference pairs per item, by coupling each item with another from the same class ($m = 100$ for IMAGENET-1K and $m = 2$ for VOXCELEB2). We make use of a validation set for model selection and hyperparameter tuning. For REPLIQA, we use the first four splits for training; from the final split, we randomly sample 10% of the data for validation and use the remaining samples for testing. For datasets with a validation set but no test partition, we use the validation set for testing and create a validation set by holding out 10% of randomly selected training samples. Then, for training RETREEVER and the baselines we make use of the training query-reference pairs, and for testing, we build the index with all the items from training, validation and test splits.

### 4.2. Baselines

Across all considered input modalities (text, image, and audio), we evaluate RETREEVER against retrieval baselines based on standard encoders and coarse-to-fine representations. For text retrieval, we further include comparisons with hierarchical approaches.

**ENCODER.** For text retrieval, we compare against DistilBERT finetuned on MSMarco[3] (Reimers and Gurevych, 2019); for image retrieval, the Visual Transformer Di-

nov2[4] (Oquab et al., 2023); for speech retrieval, the Audio Spectrogram Transformer (AST) model fine-tuned on AudioSet[5] (Gong et al., 2021b;a). Additionally, we compare with dataset-specific versions of these encoders (**Encoder-MLP**), obtained by attaching a multilayer perceptron (MLP) projection head to each encoder and fine-tuning the resulting model separately on each dataset. The MLP layer has input and output dimensions equal to the embedding size, and is trained using the contrastive loss in Eq. (1), with cosine similarity as the similarity measure. We report a sensitivity analysis with respect to the choice of encoder in Appendix E and Appendix F.

**ENCODER-MRL.** We introduce a coarse-to-fine hierarchy into encoder embeddings by training the projection layer with the contrastive loss in a Matryoshka Representation Learning fashion (Kusupati et al., 2022) (see Appendix B). This training induces nested semantic granularity in the embedding space, making Encoder-MRL directly comparable to RETREEVER's multi-level representations.

**Hierarchical clustering.** For text retrieval, we adapt k-means (McQueen, 1967) and GMM (McLachlan et al., 2019) to perform a top-down hierarchical k-means (**HIER-KMEANS**) or GMM (**HIER-GMM**) clustering, where documents are iteratively partitioned into two clusters at each level of the hierarchy based on the cosine similarity of their embeddings as extracted by an encoder. This recursive process continues until the hierarchy reaches the same depth of RETREEVER. A similar procedure is applied for building the index: contexts are greedily routed through the learned tree and assigned to the leaf they reach. We apply tree search for both HIER-KMEANS and HIER-GMM, where instead of greedily traversing the tree (which would give poor performance), we compute a global search to find the leaf with the highest score: all split-node clustering models are evaluated, then their scores are propagated and aggregated top-down. Finally, the contexts assigned to the selected leaf are re-ranked based on the cosine similarity between query and context embeddings.

**RAPTOR** (Sarthi et al., 2024) is a hierarchical method that

---

[2]*sentence-transformers/reddit-title-body*
[3]*sentence-transformers/msmarco-distilbert-cos-v5*

[4]*facebook/dinov2-large*
[5]*MIT/ast-finetuned-audioset-10-10-0.4593*

*Table 2.* Comparison of similarity search-based methods using fine representations. We measure retrieval performance with RE-CALL@10 (%) and NDCG@10 (%), with best results per metric and dataset in bold. For each dataset, we use an encoder appropriate to its data modality, as described in Section 4.2

| MODEL | NQ | | IMAGENET1K | | VOXCELEB2 | |
|---|---|---|---|---|---|---|
| | REC@10↑ | NDCG@10↑ | REC@10↑ | NDCG@10↑ | REC@10↑ | NDCG@10↑ |
| ENCODER | 57.53 | 42.15 | 92.89 | 73.60 | 85.65 | 44.94 |
| ENCODER + LINEAR | 68.61 | 47.62 | 93.84 | 76.39 | 96.68 | 75.02 |
| ENCODER + MLP | 75.10 | 52.23 | 93.49 | 79.66 | 96.25 | 73.38 |
| ENCODER + MRL | 67.10 | 46.60 | 93.87 | 76.29 | 95.61 | 70.27 |
| RETREEVER - STOCHASTIC | 77.36 | 54.48 | 94.34 | 78.82 | 99.31 | 92.60 |
| RETREEVER | **77.50** | **54.58** | **94.41** | **79.68** | **99.43** | **93.48** |

chunks long documents into shorter segments and then recursively builds a bottom-up tree via clustering and LLM-based summarization. Although designed for document QA, its core mechanism is directly relevant here: it recursively clusters and summarizes embedded text chunks into a hierarchy, building a tree structure of contexts. This makes it a highly relevant baseline for our retrieval datasets, where each context can be viewed as one of those smaller segments. For a fair assessment of its retrieval accuracy, we apply a similar tree search as for hierarchical clustering, by scoring nodes using the cosine similarity between query and summary embeddings. The contexts assigned to the selected leaf are then reranked again via the cosine similarity. Note that this is the only method we fit on the test corpus, as it cannot be applied to unseen documents and does not scale to our training sets.

### 4.3. Implementation Details

For all methods, we use representation size of 1024 and, for text retrieval, 512-token input truncation. RE-TREEVER has depth 10 and uses 16 attention heads for the cross-attention split. In RETREEVER, level embeddings attend to frozen encoder tokens with globally shared $W_q, W_k, W_v$, followed by node-specific linear readouts; this adds 3.7M trainable parameters. Median training times were 8.89h/7.88h/3.44h/6.00h on NQ/HotpotQA/RepLiQA/TopiOCQA using 1 H100 80GB GPU, and 3.65h/13.10h on ImageNet-1K/VoxCeleb2 using 4 H100 GPUs, with VoxCeleb2 slower due to on-the-fly audio feature computations. When not specified otherwise, we use exact search with cosine similarity (ENCODER) or nTVD (RETREEVER) for top-$k$ retrieval. For coarse-to-fine comparison, we use node assignments at level $h$ for RETREEVER, and the first $2^h$ dimensions of Encoder embeddings to compare at the same representation size. See Appendix B for exhaustive details.

### 4.4. Experimental Results

**Comparison with tree based methods** Table 1 compares tree-based methods' retrieval performance and inference time, across all text datasets. Remarkably, RETREEVER

achieves the best retrieval accuracy while having a runtime several order faster than Raptor's, and similar to the other hierarchical retrieval methods. On average, RETREEVER processes each query in just 10 ms. Compared to RAPTOR, it is about $160\times$ faster while obtaining performance boosts of around 30%, despite not making LLM calls for building the tree. Finally, RETREEVER is about $2\times$ slower than HIER-KMEANS and HIER-GMM but achieves at least $2.5\times$ better performance. The increased latency is due to our model having higher capacity split functions, as compared to dot products for computing cosine similarity. Overall, these results provide strong evidence for learning tree structures optimal for retrieval.

**Fine Representation Comparison** Table 2 reports retrieval performance across modalities for all representation-based methods. We observe similar results on the other text datasets in Table 7. First, RETREEVER consistently achieves the best results across both recall and nDCG, substantially outperforming ENCODER. This demonstrates that learning a tree does not compromise the encoder's representational power, and that structual interpretability does not come at the cost of retrieval performance. Second, comparing ENCODER with other baselines shows that finetuning the encoder on task-specific data unsurprisingly improves performance. The superior results of our finetuning strategy can be attributed to its use of higher-capacity split functions (attention-based versus linear or MLP adaptors). Finally, we observe that explicitly training for high-quality coarse representations (ENCODER-MRL and RETREEVER-STOCHASTIC) slightly degrades fine-level representation power. We further investigate the impact of coarse-to-fine optimization strategies in what follows.

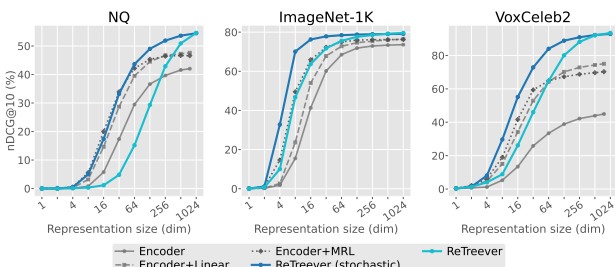

*Figure 3.* NDCG@10 (%) vs. representation size across modalities, using a single index. For each dataset, we use an encoder appropriate to its data modality, as described in Section 4.2.

**Coarse-to-fine Representation** In Figure 3, we compare retrieval performance across text, image and speech datasets, depending on the representation size. We observe that RE-TREEVER-STOCHASTIC delivers the strongest coarse representations, as measured by NDCG@10, consistently across modalities, improving over all baselines at any representation size. In particular, it outperforms ENCODER-MRL, the other method explicitly optimizing for high quality coarse

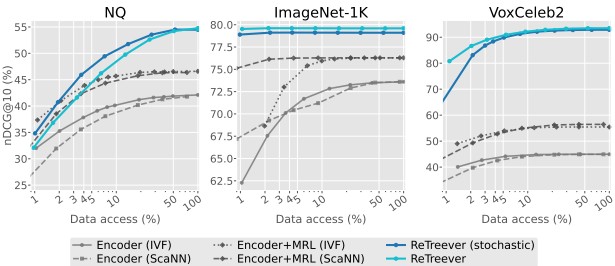

*Figure 4.* NDCG@10 (%) vs (%) Data Accessed across modalities. RETREEVER is evaluated using leaf-level indices, while baselines make use of the approximate searches: IVF and SCANN. For each dataset, we use an encoder appropriate to its data modality, as described in Section 4.2.

representations. Strong coarse representations cut inference cost massively by 75% while retaining 95% of NDCG@10 (averaged across all datasets) (see Appendix G, Figure 12). Finally, imposing coarse-to-fine constraints degrades performance of the finer representations across all datasets, as seen when moving from ENCODER-LINEAR to ENCODER-MRL, and slightly from RETREEVER to RETREEVER-STOCHASTIC (see Table 2).

**Approximate Data Search** In Figure 4, we study retrieval performance as we progressively restrict the amount of data that can be accessed during similarity search. On all datasets, we observe significant performance degradation for all methods with data budgets below 30%, with the exception of IMAGENET-1K where RETREEVER and EN-CODER+MRL (ScaNN) prove to be robust to extremely low data regimes. In general, our tree-based approach consistently outperforms ENCODER and ENCODER-MRL combined with standard approximate search methods across all budgets. Remarkably, RETREEVER performs well even though it is not designed as an approximate search method. Unlike IVF and ScaNN (Guo et al., 2020), which use task-agnostic clustering to create partitions, our method learns hierarchical partitions directly optimized for retrieval. This is further evidence that our method learns a meaningful organization of the data. Finally, we observe that RETREEVER and RETREEVER-STOCHASTIC perform similarly over all data access budgets, except on VOXCELEB2 where the stochastic variant is substantially stronger at low data-access regimes. Overall, these results demonstrate that leveraging the tree's partitions achieves at least $2\times$ latency reduction with minimal impact on retrieval quality.

We further evaluate a complementary setting in Figure 5, where the search within each selected leaf is performed with HNSW (Malkov and Yashunin, 2018), one of the most widely used approximate nearest-neighbor methods. This highlights that RETREEVER's multi-index design is orthogonal to the choice of within-leaf search backend: the tree first

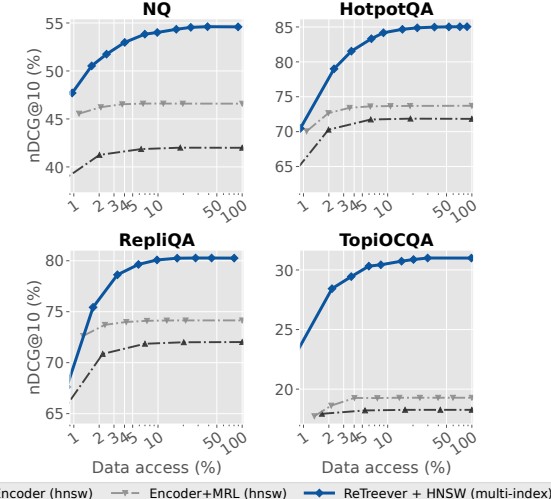

*Figure 5.* NDCG@10 (%) vs. data accessed (%) on the four text datasets. We compare RETREEVER multi-index hybrid variant that partitions documents using level-5 tree assignments, giving 32 buckets, and applies HNSW within the selected buckets with the baselines that run HNSW directly on ENCODER or ENCODER+MRL representations. The results show that RE-TREEVER's learned tree indexing is complementary to the choice of within-partition search backend.

routes the query to relevant partitions, and each partition can then be searched using either an exact or approximate index. Combining tree-based partitioning with HNSW inside the leaves outperforms HNSW built directly over ENCODER or ENCODER+MRL representations across the four text datasets, showing that the learned hierarchy complements a strong ANN backend. This flexibility becomes especially important at larger corpus sizes; in Appendix I, we stress-test RETREEVER on the 124.5M-context Reddit Title–Body corpus and show that global ANN construction can become memory-limited, while the leaf-level multi-index remains practical under the same memory budget.

## 5. Inspecting the Tree

By learning node assignment probabilities, RETREEVER has the appealing side-effect of providing an organization of reference documents, which serves as an interface for inspecting the model and analyzing the contexts. In this section, we probe RETREEVER's organization via several tools and report evidence that semantic groupings naturally emerge when optimizing for retrieval. We release an anonymous interactive visualization website that renders trees learned by RETREEVER on text and image datasets for inspection; see Appendix M for details and examples.

**Tree Congruence** We first examine whether RETREEVER learns representations that are congruent with its tree structure, despite being trained solely on query-context alignment labels. In Figure 6, we provide quantitative evidence that

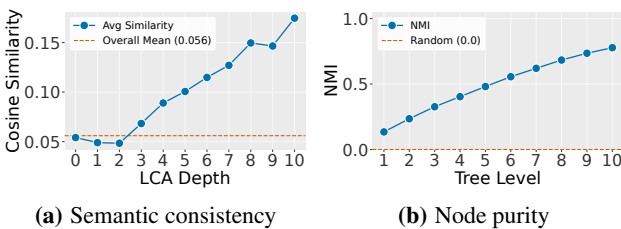

**(a)** Semantic consistency     **(b)** Node purity

*Figure 6.* Learned trees exhibit semantic structure. (a) Hierarchical Semantic Consistency: Documents with deeper common ancestors are more semantically similar (NQ). (b) Class-Node separation across tree levels: Node purity improves with tree depth, as measured by normalized mutual information (NMI), which quantifies how much information tree-level node assignments contain about class labels, normalized to lie between 0 and 1 (IMAGENET-1K).

RETREEVER learns trees that exhibit semantic structure across modalities. For NQ, we study whether documents assigned to nearby nodes are more similar, on average, than documents assigned to different subtrees. We compute the cosine similarity between document embeddings (extracted with DistilBERT) grouped by the depth of their *lowest common ancestor* (LCA), the deepest node in the tree that is an ancestor of both contexts. The hypothesis is that contexts routed to different nodes at earlier layers (i.e. with a shallow LCA closer to the root) should be thematically more distinct than those separated deeper in the tree. The strong positive correlation shows that documents sharing deeper (i.e. closer to the leaves) common ancestors are semantically more similar. For IMAGENET-1K, we study whether images from the same class are routed similarly through the tree. Given the tree paths of the images, we compute the *Normalized Mutual Information* (NMI) between their class distribution and their tree level assignments. NMI increases with level depth, indicating that class purity increases with depth and demonstrating progressively finer semantic separation. See Appendix H for an extended analysis.

**Topics and Keywords Coherence**  To further inspect the learned structure in a human-understandable way, we resort to topic models for extracting and analyzing the distribution of topics and keywords of the contexts assigned to nodes across the tree. We present a visualization of the RETREEVER tree trained on NQ, highlighting the topics and keywords associated with several nodes (Figure 2). For each node $t$, we collect all contexts assigned to the leaves of the subtree rooted at $t$ and extract topics and keywords using the method from Kilgarriff (2009). A quick inspection of these keywords reveals that the learned hierarchy reflects the semantic relationships among contexts. For example, the node whose contexts are on the topic of *media* (node 5) has child nodes focused on *publishing* and *TV*. Furthermore, the path from node 5 to node 54 illustrates a consistent refinement of topics and keywords, progressing from *Media* down to *Television seasons*. Similarly Figure 7 shows a tree path

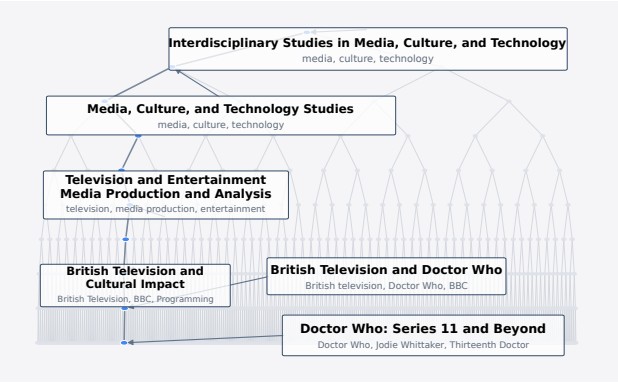

*Figure 7.* Example path in a ReTreever tree learned on NQ, showing semantic refinement from broad media-related topics to a fine-grained Doctor Who subtree.

that shows semantic refinement from *Media* to *television* to *British Television* and finally *Doctor Who*.

## 6. Discussion

This work presents RETREEVER, a hierarchical retriever system that generates tree-based representations optimized for retrieval. By aligning coarse-to-fine representations with a learned hierarchy, RETREEVER enables flexible trade-offs between representation size, inference latency, and retrieval accuracy, while remaining competitive with flat retrieval methods and offering substantially improved efficiency at low data-access regimes. Beyond efficiency, the learned tree provides an inspectable organization of the corpus, allowing practitioners to examine intermediate groupings, and offering some degree of post-hoc structural interpretability that is largely absent from standard embedding-based retrieval systems. Several promising directions emerge from this work. While RETREEVER currently operates on top of a frozen encoder and a single learned hierarchy per dataset, extending the framework to support adaptive or incremental tree updates could improve its applicability to dynamic or evolving corpora. Learning variable-depth or variable-width trees may further improve efficiency by allocating capacity where it is most needed, while tighter integration between encoder and tree learning could strengthen semantic alignment.

## Impact Statement

This paper presents work whose goal is to advance the field of Machine Learning. There are many potential societal consequences of our work, none of which we feel must be specifically highlighted here. We however believe that contributing towards transparent and verifiable retrievers should benefit society in the long term, and is essential in fostering trust and accountability of retrieval systems.

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

# A. RETREEVER's Additional Design Options

RETREEVER can be instantiated in many different ways. In this section we report a selection of the design choices we explored in our experiments and that are implemented in our codebase.

## A.1. Split Functions

A split function $s_{\theta_t} : \mathcal{X} \to [0,1]$ determines the routing probability of an input $\mathbf{x}_i \in \mathcal{X}$ through node $t$. The primary requirement for a split function is that it outputs a scalar $\in \mathbb{R}$, based on which we compute the probabilities of routing the input to node $t$'s left or right children $t_{\text{left}}$ and $t_{\text{right}}$ (see Appendix J). Below, we describe different types of split functions that can be used.

### A.1.1. LINEAR SPLIT FUNCTION

The simplest form of a split function is a linear projection, similar to the approach used in (Zantedeschi et al., 2021). For a given split node $t$, with left and right children $t_{\text{left}}$ and $t_{\text{right}}$, the split function is defined as:

$$s_{\theta_t}(\mathbf{x}_i) = \theta_t^\top \mathbf{x}_i, \tag{2}$$

where $\theta_t \in \mathcal{X}$ represents a learnable hyperplane. Each split node learns a separate hyperplane, and there are no shared parameters across the tree.

### A.1.2. MLP SPLIT FUNCTION

A more expressive alternative is to use a learnable neural network, modeled as a Multi-Layer Perceptron (MLP). This MLP $S_\Theta : \mathcal{X} \to \mathbb{R}^{|\mathcal{T}_B|}$ maps an input $\mathbf{x}_i$ to a routing probability for each branching node in the tree, where $\mathcal{T}_B$ is the set of branching nodes (i.e. non leaf nodes). The MLP consists of multiple layers with nonlinear activations, such as ReLU, and incorporates dropout for regularization. Unlike the linear split function, which maintains separate parameters per node, the MLP split function shares parameters across different nodes while still allowing for node-specific learning.

### A.1.3. CROSS-ATTENTION SPLIT FUNCTION

While the linear and MLP split functions operate on dense passage embeddings for the entire document, the cross-attention split function allows us to leverage token-level embeddings for more expressive routing. This method utilizes a cross-attention mechanism between learnable node embeddings and text tokens to determine the routing probabilities at each split node.

Let the input text $\mathbf{x}_i \in \mathbb{R}^{n_d \times d_{\text{emb}}}$ consist of $n_d$ embedded tokens, encoded by the encoder $E$. Instead of a simple projection, we introduce learnable level embeddings $\mathbf{e}_l \in \mathbb{R}^{n_e \times d'_{\text{emb}}}$ for each tree level $l$. These level embeddings interact with the text tokens via a cross-attention mechanism, where the level embeddings serve as queries, and the text tokens provide keys and values.

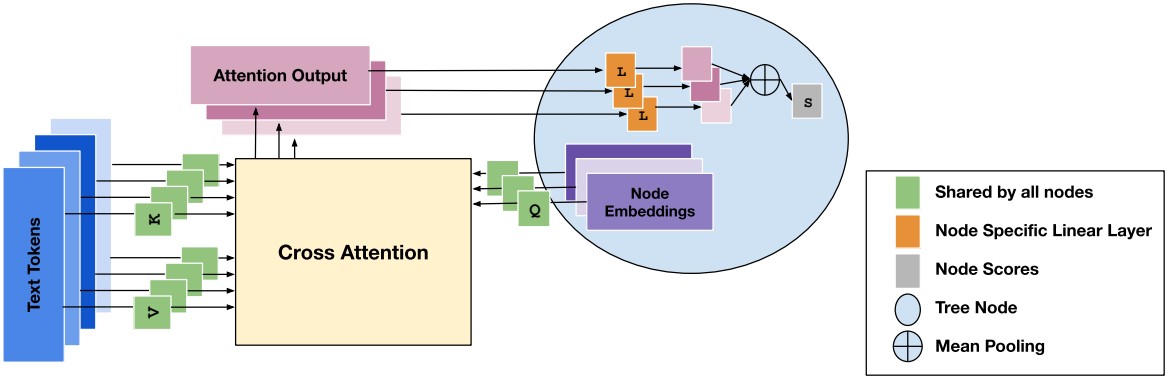

*Figure 8.* RETREEVER's cross-attention split function with node scoring done by a per node linear map followed by a mean of scores.

We define the attention mechanism as follows:

$$\text{Attention}(\mathbf{Q}, \mathbf{K}, \mathbf{V}) = \text{softmax}\left(\frac{\mathbf{Q}\mathbf{K}^\top}{\sqrt{d_k}}\right)\mathbf{V}, \tag{3}$$

where:

$$\mathbf{Q} = \mathbf{e}_l \mathbf{W}_q^\top, \tag{4}$$
$$\mathbf{K} = \mathbf{x}_i \mathbf{W}_k^\top, \tag{5}$$
$$\mathbf{V} = \mathbf{x}_i \mathbf{W}_v^\top. \tag{6}$$

Here, $\mathbf{W}_q \in \mathbb{R}^{d_k \times d'_{\text{emb}}}$, $\mathbf{W}_k \in \mathbb{R}^{d_k \times d_{\text{emb}}}$, and $\mathbf{W}_v \in \mathbb{R}^{d_k \times d_{\text{emb}}}$ are learnable projection matrices shared across the tree, while $d_k$ represents the dimension of the projected queries and keys. This formulation describes a single-head attention mechanism but can be naturally extended to multi-head attention by introducing independent projection matrices for each head and concatenating the resulting outputs.

**Parameter sharing and node-specific scoring.** The learnable level embeddings act as depth-specific queries over the frozen encoder token representations. For each level $\ell$, their contextualized outputs form a shared level representation, which is used by all nodes at that level. Node specificity is introduced only at the final scoring stage: each node $t$ at level $\ell$ has its own learned linear readout that maps the shared level representation to a scalar routing score. When multiple level embeddings are used, we apply the same node-specific readout to each contextualized embedding and average the resulting scores, equivalently mean-pooling the contextualized outputs before the linear readout. This aggregation is shown in Figure 8. For the configuration reported in Table 12, RETREEVER adds 3.7M trainable parameters on top of the frozen encoder.

We compare the effect of different split functions on retrieval performance across various datasets in Table 3. Notably, the cross-attention split function achieves the best retrieval metrics across all datasets, while the MLP split function worse than the linear one. This surprising result can be explained by highlighting that both cross-attention and linear splits learn virtual embeddings (the level embeddings combined with node projection functions in the cross-attention and the hyper-plane in the linear one) and compare them with the input. Further investigation would be required to confirm this speculation.

*Table 3.* Effect of split-function choices on full-representation retrieval RECALL@10 (%) and NDCG@10 (%) across three datasets, with best results per metric and dataset in bold.

| SPLIT FUNCTION | NQ | | IMAGENET1K | | VOXCELEB2 | |
|---|---|---|---|---|---|---|
| | REC@10 ↑ | NDCG@10 ↑ | REC@10 ↑ | NDCG@10 ↑ | REC@10 ↑ | NDCG@10 ↑ |
| LINEAR | 69.71 | 48.60 | 93.58 | 78.43 | 96.17 | 71.58 |
| MLP | 57.69 | 39.06 | 92.85 | 75.60 | 82.17 | 39.21 |
| CROSS-ATTENTION | **77.50** | **54.58** | **93.92** | **79.59** | **99.43** | **93.48** |

## A.2. Discussion on Cross-Attention Split Function

**Use of level embeddings as queries** The cross-attention split function establishes a structural parallel to the encoder-decoder framework found in standard Transformer architectures. In this analogy, the text encoder provides the keys and values, while the learnable level embeddings $e_l$ function analogously to decoder queries. Rather than generating text, these embeddings act as learnable semantic probes that query the document to extract features relevant to hierarchical routing. We believe that by using the same level embeddings across the entire corpus during training, the model effectively learns to store a "global memory" of the dataset's distribution within the tree parameters. We hypothesize that this allows the routing mechanism to perform a content-dependent summarization of the document, thereby asking which token-level features correspond to the global concepts associated with that specific depth, before specializing these summaries into node-specific scores through a final linear projection. This separation of concerns allows the model to maintain a shared semantic context at each level, while retaining the granularity required for individual routing decisions. Further investigation is required to confirm our hypothesis.

### A.3. Number of Heads in The Cross-Attention Mechanism

We show in Appendix A.1.3 that the cross-attention split function naturally extends to a multi-head setting. Here, we evaluate different numbers of attention heads across datasets. As shown in Table 4, 16 heads achieve the best performance on 3 of the 6 dataset–metric combinations. We therefore use 16 heads in all experiments.

*Table 4.* Effect of the number of cross-attention heads on full-representation retrieval RECALL@10 (%) and NDCG@10 (%) across three datasets, with best results per metric and dataset in bold.

| # HEADS | NQ | | IMAGENET1K | | VOXCELEB2 | |
|---|---|---|---|---|---|---|
| | REC@10↑ | NDCG@10↑ | REC@10↑ | NDCG@10↑ | REC@10↑ | NDCG@10↑ |
| 2 | 73.14 | 50.09 | 93.85 | 79.02 | 99.26 | 92.23 |
| 4 | 73.73 | 50.93 | **94.02** | 79.98 | 99.29 | 92.25 |
| 8 | 74.43 | 51.22 | 93.93 | 80.51 | 99.37 | 92.85 |
| 16 | **77.50** | **54.58** | 93.92 | 79.59 | **99.43** | **93.48** |
| 32 | 75.01 | 51.69 | 93.82 | **80.59** | 99.27 | 92.07 |

### A.4. Choice of Tree Depth

We also study the effect of tree depth on retrieval performance in Table 5. Increasing depth generally improves performance, and the largest depth we evaluate (12) achieves the best results for most dataset–metric combinations. However, on VOXCELEB2 we observe signs of overfitting at depth 12, where shallower trees (depth 8 or 10) perform better. Based on this tradeoff, and to keep the finest-level representation size comparable to standard encoders, we use depth 10 in our experiments, yielding leaf-level representations of length 1024.

*Table 5.* Effect of tree depth on full-representation retrieval RECALL@10 (%) and NDCG@10 (%) across three datasets, with best results per metric and dataset in bold.

| DEPTH | NQ | | IMAGENET1K | | VOXCELEB2 | |
|---|---|---|---|---|---|---|
| | REC@10↑ | NDCG@10↑ | REC@10↑ | NDCG@10↑ | REC@10↑ | NDCG@10↑ |
| 6 | 64.51 | 42.98 | 94.06 | 77.84 | 98.54 | 86.14 |
| 8 | 72.72 | 50.19 | **94.08** | 79.63 | 99.27 | **92.36** |
| 10 | 73.81 | 50.85 | 94.01 | 80.62 | **99.31** | 92.27 |
| 12 | **75.55** | **52.30** | 93.69 | **81.14** | 99.07 | 89.60 |

### A.5. Number of Level Embeddings

We also ablate the number of learnable level embeddings per tree level. As described in Appendix A.1.3, our cross-attention split function maintains $n_\ell$ learnable embeddings at each level; for a depth-10 tree this corresponds to $10n_\ell$ level embeddings in total. These embeddings are contextualized via cross-attention with the input, and then mapped to node scores through node-specific projection functions (averaging scores when multiple level embeddings are used). Table 6 shows that increasing $n_\ell$ generally improves retrieval performance across datasets, but with diminishing returns beyond a small number of embeddings. Based on this tradeoff, we use $n_\ell = 8$ embeddings per level in all experiments.

*Table 6.* Effect of the number of embeddings per level on full-representation retrieval RECALL@10 (%) and NDCG@10 (%) across three datasets, with best results per metric and dataset in bold.

| # EMBEDDINGS / LEVEL | NQ | | IMAGENET1K | | VOXCELEB2 | |
|---|---|---|---|---|---|---|
| | REC@10↑ | NDCG@10↑ | REC@10↑ | NDCG@10↑ | REC@10↑ | NDCG@10↑ |
| 1 | 73.71 | 50.73 | 93.95 | 78.23 | 98.31 | 92.29 |
| 2 | 73.69 | 50.56 | 93.91 | 78.19 | 98.98 | 92.05 |
| 4 | **74.67** | 51.39 | **94.03** | 79.28 | 99.27 | 92.66 |
| 8 | 74.49 | **51.46** | 93.87 | **80.95** | **99.33** | **92.79** |

# B. Coarse-to-Fine Representations Across Datasets

Figure 9 compares coarse-to-fine representation quality by reporting NDCG@10 as a function of representation size. Shallower tree levels yield shorter, coarser representations, enabling faster retrieval at the cost of reduced NDCG, thus inducing an explicit latency–quality tradeoff. RETREEVER-STOCHASTIC is trained to strengthen these intermediate-level representations, and we observe this consistently across datasets: at coarse and intermediate dimensions, RETREEVER-STOCHASTIC outperforms RETREEVER across nearly all sizes, while RETREEVER edges ahead at the finest level, as expected when additional stochastic constraints trade a small amount of peak performance for improved coarse representations.

For baselines, we obtain coarse representations from the base encoder by truncating the embedding to its first $k$ dimensions. Since the base encoder is not tuned for our retrieval tasks, we also include an *Encoder+Linear* adapter baseline, which is conceptually similar to RETREEVER as a lightweight adaptation layer on top of a frozen encoder. Finally, we compare against Matryoshka Representation Learning (MRL), which explicitly trains embeddings to be retrieval-effective under truncation. Across 5 of the 6 datasets shown, RETREEVER-STOCHASTIC provides the best NDCG@10 on most coarse granularities, indicating that our gains come not only from explicit coarse-to-fine training (as in MRL) but also from the hierarchical tree structure that induces task-aligned semantic groupings. On REPLIQA, we observe a different behavior: the probability-bounding constraints can hurt performance at the coarsest levels, whereas relaxing them improves coarse representations; we analyze this effect further in Appendix J.

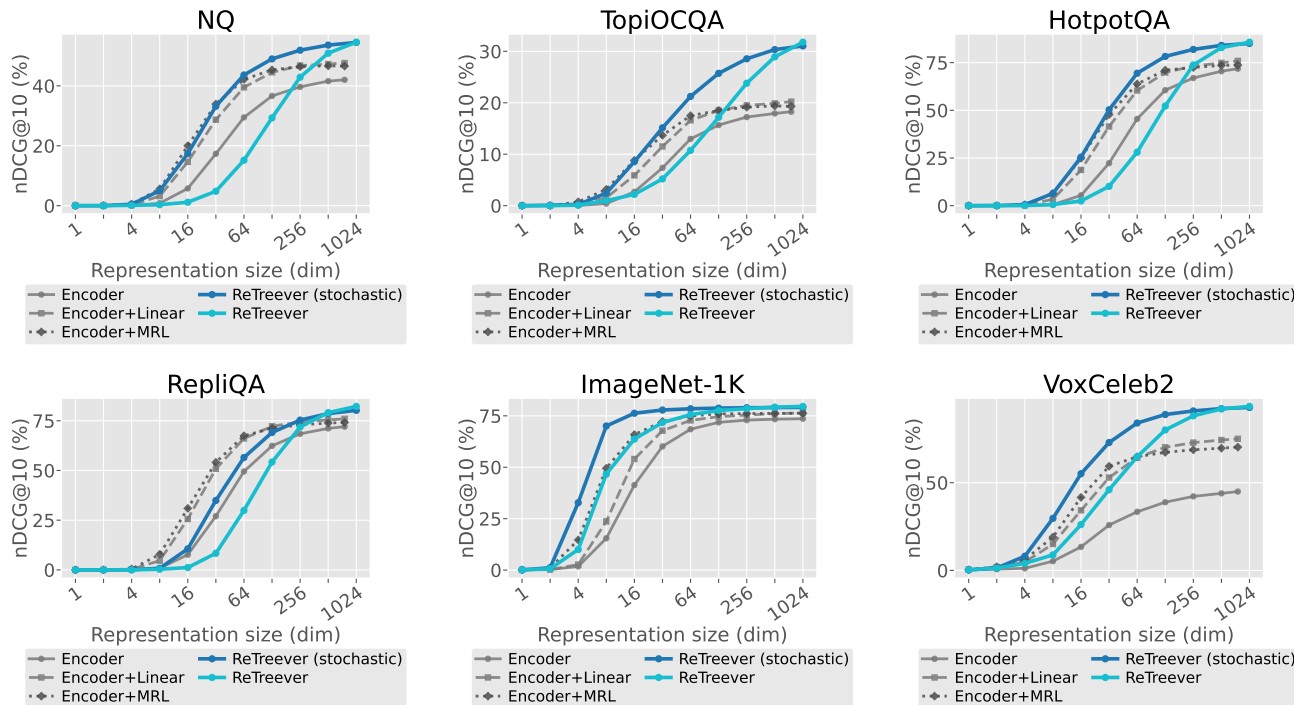

*Figure 9.* Coarse-to-Fine Representations Across Datasets

## C. Multi-index Performance Across Datasets

As described in Section 3.3, we consider two indexing strategies for RETREEVER. In *single-index retrieval* (see Appendix B), the probability vector produced by routing a text item through the tree to a given tree level is treated as a dense representation. We build a single FAISS-style index over these representations for the entire corpus, and retrieve by mapping queries into the same representation space and performing nearest-neighbor search.

In *multi-index retrieval*, we instead exploit the fact that RETREEVER outputs a distribution over tree nodes, which can be interpreted as a distribution over latent, hierarchical concepts induced by the learned tree. We use these latent concepts to cluster documents and restrict search to only the most relevant clusters. Concretely, at level 10 of a depth-10 tree there are $2^{10} = 1024$ leaf nodes; we assign each document to the leaf with maximum assignment probability (i.e., the *argmax* of its leaf distribution), and build a separate index per leaf cluster. At query time, we route the query through the same tree, select its top-$k$ leaf assignments, and search only the corresponding indices. This can substantially reduce the number of documents examined, with parameter $k$ allowing us trade off accuracy for latency.

We report *data access percentage* as a proxy for latency, since retrieval time is approximately linear in the number of documents searched in this setup. This clustering-based retrieval is conceptually related to standard approximate search methods such as IVF and ScaNN, which we include as baselines.

Figure 10 compares these approaches across six datasets. We evaluate IVF and ScaNN built on the base encoder representations. Since the base encoder is not tuned to these retrieval tasks, we also include IVF and ScaNN on top of an ENCODER+MRL baseline, which explicitly trains coarse-to-fine representations and is therefore the closest non-tree analogue to our setting.

Overall, RETREEVER provides a favorable quality–latency tradeoff. On TOPIOCQA, VOXCELEB2, and IMAGENET1K, RETREEVER outperforms the approximate-search baselines at every access budget, and on NQ it is better at nearly all budgets. On HOTPOTQA and REPLIQA, RETREEVER trails the baselines at very small access budgets, but becomes competitive around the $5\%$ and $10\%$ access points respectively, still offering substantial latency reductions (often by more than an order of magnitude) with limited loss in retrieval quality.

We hypothesize that HOTPOTQA and REPLIQA induce more diffuse concept structure, requiring a broader set of leaves to cover the relevant evidence. This is particularly plausible for HOTPOTQA, where multiple supporting passages are combined into a single positive context, increasing the number of distinct concepts that must be retrieved jointly. In such cases, deeper trees (i.e., more leaves) should allow finer partitions and improve the multi-index tradeoff, which we leave as a promising direction for future work.

We further hypothesize that TOPIOCQA, IMAGENET1K, and VOXCELEB2 yield sharper hierarchical organization because their queries are more informative and constrain the relevant evidence more strongly. In TOPIOCQA, queries include conversational history, which typically disambiguates entities and intent and reduces the set of plausible matching contexts. In IMAGENET1K and VOXCELEB2, queries and contexts are drawn from the same modality and distribution (image-to-image and audio-to-audio retrieval, respectively), so relevance is expressed through aligned low-level and mid-level features rather than a cross-distribution semantic mapping. In these settings, the tree can learn high-purity partitions (see Figure 15), concentrating probability mass on a small number of leaves and making multi-index retrieval efficient.

## D. DistilBERT Over All Datasets

Table 7 reports RECALL@10 and NDCG@10 on the four text retrieval datasets (NQ, HOTPOTQA, REPLIQA, and TOPIOCQA) using `sentence-transformers/msmarco-distilbert-cos-v5` as the base encoder. We evaluate finest-level retrieval using (i) the base encoder's dense representations and (ii) RETREEVER's leaf assignment distributions treated as representations, for both RETREEVER and RETREEVER-STOCHASTIC.

Since the base encoder is not tuned to these downstream datasets, we additionally compare against adapter baselines that fine-tune the encoder with either a linear projection (ENCODER+LINEAR) or a two-layer MLP with an up-projection and down-projection with nonlinearity and dropout (ENCODER+MLP). Finally, we include ENCODER+MRL (Matryoshka Representation Learning), which explicitly trains coarse-to-fine representations and is therefore the closest non-tree analogue to our setting.

Across all datasets, adapting the encoder with a linear or MLP head consistently improves both RECALL@10 and

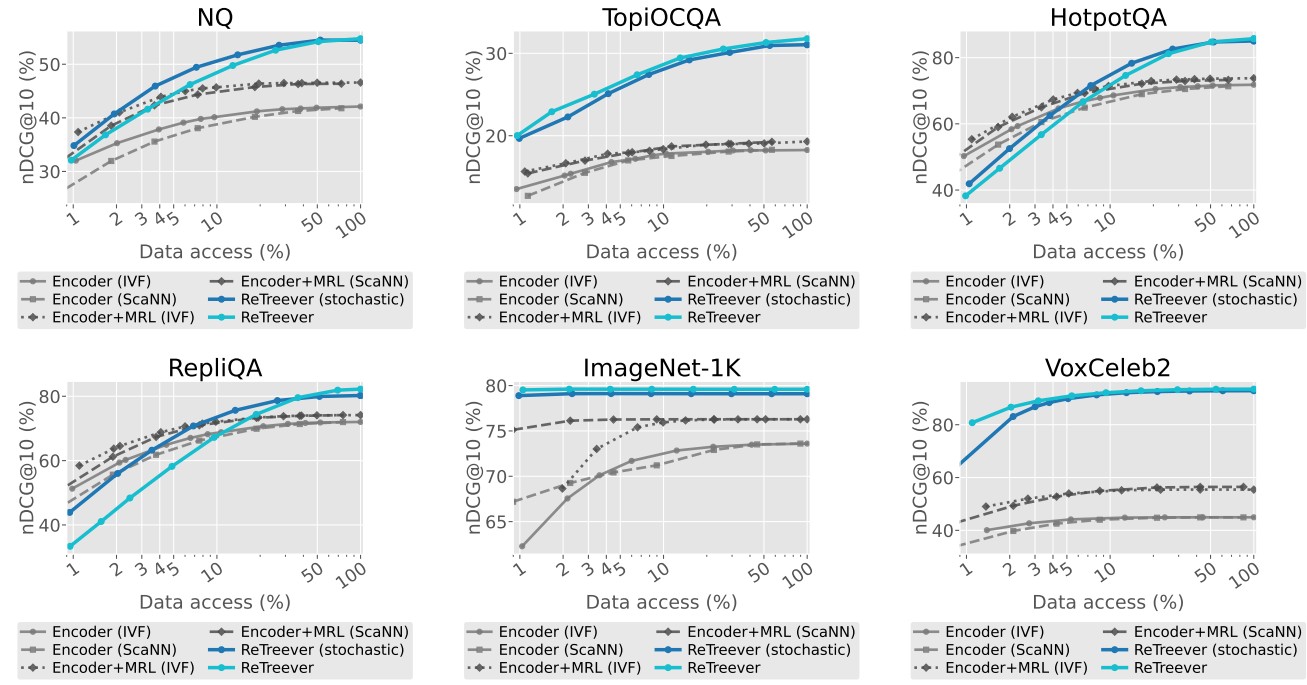

*Figure 10.* Multi index Results over all datasets

NDCG@10 over the base encoder. In contrast, ENCODER+MRL yields weaker finest-level performance, consistent with the general tradeoff that enforcing coarse-to-fine structure can reduce peak performance at the finest granularity. We observe a similar pattern for RETREEVER: at the finest level, RETREEVER is consistently stronger than RETREEVER-STOCHASTIC, while RETREEVER-STOCHASTIC provides substantially better coarse representations across levels (see Figure 9). Overall, both RETREEVER variants outperform all baselines on all datasets under the finest-level RECALL@10 and NDCG@10 metrics in Table 7.

*Table 7.* Full representation retrieval RECALL@10 (%) and NDCG@10 (%), with best results per metric and dataset in bold, on the text retrieval datasets.

| MODEL | NQ | | HOTPOTQA | | TOPIOCQA | | REPLIQA | |
|---|---|---|---|---|---|---|---|---|
| | REC@10↑ | NDCG@10↑ | REC@10↑ | NDCG@10↑ | REC@10↑ | NDCG@10↑ | REC@10↑ | NDCG@10↑ |
| ENCODER | 57.53 | 42.15 | 82.14 | 71.82 | 31.03 | 18.26 | 82.39 | 72.02 |
| ENCODER + LINEAR | 68.61 | 47.62 | 86.51 | 75.90 | 36.86 | 20.20 | 86.50 | 75.96 |
| ENCODER + MLP | 75.10 | 52.23 | 87.72 | 76.88 | 36.24 | 20.63 | 85.66 | 74.99 |
| ENCODER + MRL | 67.10 | 46.60 | 84.46 | 73.70 | 34.50 | 19.30 | 85.20 | 74.15 |
| RETREEVER - STOCHASTIC | 77.36 | 54.48 | 92.95 | 85.04 | 51.80 | 31.00 | 89.36 | 80.23 |
| RETREEVER | **77.50** | **54.58** | **93.47** | **85.71** | **52.11** | **31.78** | **90.84** | **82.22** |

## E. Comparing Different Text Encoders

We repeat the same set of baselines and training variants as in Table 7, but replace the base encoder with additional text encoders. We first evaluate `BAAI/bge-large-en-v1.5`, a larger retrieval-oriented encoder pre-trained on large-scale text and fine-tuned on retrieval and QA-style tasks. Results are reported in Table 8. We also evaluate Contriever (Izacard et al.), using `facebook/contriever`, as a complementary unsupervised dense retrieval backbone; results are reported in Table 9.

A notable difference relative to DistilBERT is that, when the encoder is already strongly tuned for retrieval or question-

answering, additional adapter-style fine-tuning can be less reliable and may even degrade performance. This effect is visible in Table 8: adding an MLP adapter decreases performance on REPLIQA relative to the base BGE encoder, and on HOTPOTQA a linear adapter outperforms MLP. The same sensitivity is reflected in RETREEVER results, suggesting that when the backbone already provides a specialized representation space, further adaptation must be applied more cautiously.

The Contriever results in Table 9 show a similar dependence on dataset and backbone alignment. RETREEVER improves substantially over the frozen Contriever encoder on NQ and TOPIOCQA, while the frozen encoder remains strongest on HOTPOTQA and REPLIQA. Overall, these results indicate that the benefits of the learned hierarchy are complementary to the choice of base encoder: RETREEVER provides coarse-to-fine representations, multi-index retrieval, and an explicit corpus organization, while the absolute finest-level retrieval performance depends on how well the frozen encoder is already aligned with each dataset.

*Table 8.* Full-representation retrieval NDCG@10 (%) with a BGE encoder across text retrieval datasets, comparing the base encoder, adapter baselines, and RETREEVER variants. Best results per dataset are in bold.

| MODEL | NQ NDCG@10 ↑ | HOTPOTQA NDCG@10 ↑ | TOPIOCQA NDCG@10 ↑ | REPLIQA NDCG@10 ↑ |
|---|---|---|---|---|
| ENCODER | 55.57 | 91.54 | 22.19 | 86.42 |
| ENCODER + MLP | 57.20 | 91.67 | 25.02 | 84.68 |
| ENCODER + LINEAR | 58.19 | **91.98** | 24.91 | **87.01** |
| ENCODER + MRL | 57.05 | 90.40 | 24.35 | 85.47 |
| RETREEVER + STOCHASTIC | 60.26 | 90.06 | **30.11** | 84.45 |
| RETREEVER | **60.46** | 91.05 | 27.82 | 85.74 |

*Table 9.* Full-representation retrieval NDCG@10 (%) with a Contriever encoder across text retrieval datasets, comparing the frozen encoder and RETREEVER variants. Best results per dataset are in bold.

| MODEL | NQ NDCG@10 ↑ | HOTPOTQA NDCG@10 ↑ | TOPIOCQA NDCG@10 ↑ | REPLIQA NDCG@10 ↑ |
|---|---|---|---|---|
| ENCODER | 46.25 | **87.98** | 21.25 | **83.61** |
| RETREEVER + STOCHASTIC | 55.20 | 83.02 | **23.50** | 80.56 |
| RETREEVER | **55.90** | 85.76 | 22.95 | 82.65 |

# F. Comparing Different Image Encoders

We evaluate RETREEVER with three image encoders spanning a wide range of capacity and inductive bias: (i) **ResNet-50**, a convolutional backbone with roughly 25.6M parameters and a 2048-dimensional feature representation; (ii) **DINOv2-Large** (`ViT-L/14`), a self-supervised Vision Transformer with a 1024-dimensional embedding and roughly 300M parameters; and finally (iii) **CLIP ViT-L/14** (`openai/clip-vit-large-patch14`), a vision-language model whose vision tower uses width 1024 and projects to a 768-dimensional shared embedding space (model size ≈ 427M parameters).

Across all three encoders, we consistently observe that RETREEVER yields markedly stronger representations at both coarse and fine levels, as compared to the non-tree baselines, including Matryoshka Representation Learning (MRL), which explicitly targets coarse-to-fine behavior but does not impose a hierarchical routing structure. As expected, the RETREEVER-STOCHASTIC variant produces the strongest coarse representations, as its explicitly trained to do so. Overall, these results support the view that learning a retrieval-optimized tree induces semantically meaningful hierarchical organization of the corpus, which directly translates into improved retrieval quality at various representation levels.

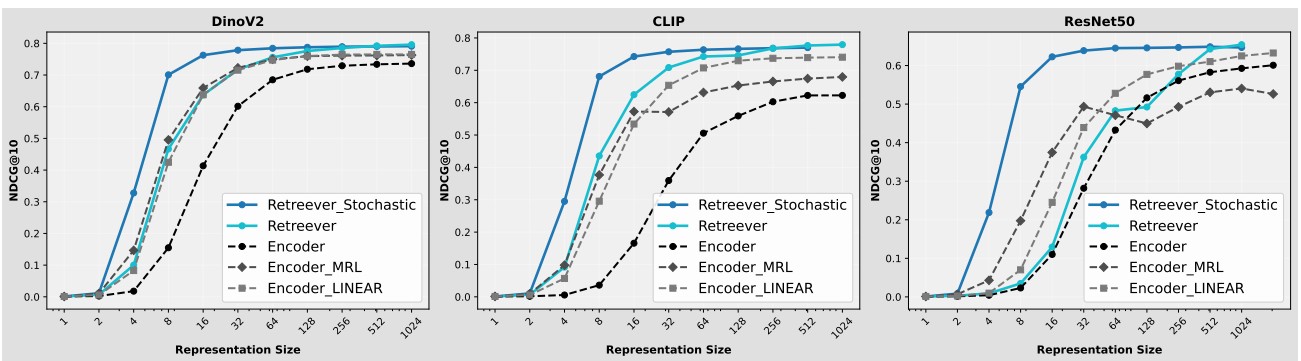

*Figure 11.* Level wise retrieval performance for three image encoders

## G. Retreever Latency vs Representation Size

We analyze the relationship between embedding size obtained from RETREEVER and its impact on inference latency and retrieval performance. As shown in Figure 12, the seconds per query (`sec/query`) increase as the representation size grows. This trend is expected, as larger representations require more computation and memory access during retrieval. We observe, that the inference latency is linearly related to the size of the representation obtained from RETREEVER. We also note from the figure that its possible to obtain a significant decrease in query latency without sacrificing on retrieval accuracy by much.

Figure 12 further illustrates the trade-off between retrieval effectiveness, measured by NDCG@10, and query latency. While larger representations significantly increase model latency, they offer only marginal improvements in retrieval performance. This provides a way to balance query latency and retrieval performance by selecting an appropriate embedding size from RETREEVER to achieve the desired trade-off.

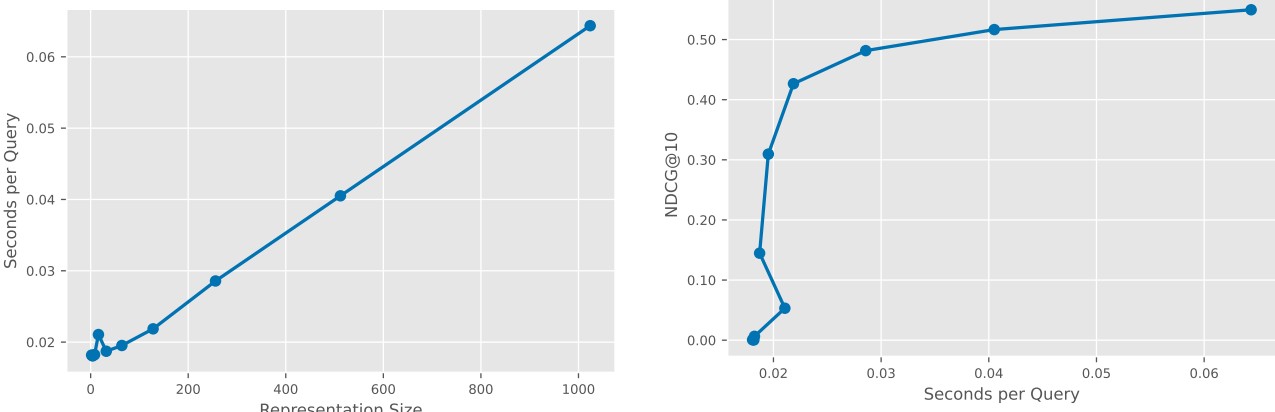

*Figure 12.* Trade-off between retrieval speed and effectiveness in Retreever. The left plot shows a linear dependence between the representation size from RETREEVER and the query latency, while the right plot demonstrates that a speedup in query inference with only a slight reduction in retrieval performance.

## H. Quantitative Metrics for Interpretability

We evaluate whether ReTreever learns a meaningful refinement hierarchy by measuring hierarchical semantic consistency. For randomly sampled pairs of documents, we assign each document to its most probable leaf (argmax) and compute the depth of the lowest common ancestor (LCA) of the two leaves. We then compute cosine similarity between the documents' original encoder embeddings and report the average similarity as a function of LCA depth. Across all four text datasets (see Figure 14, similarity increases monotonically with LCA depth, and is substantially above the global mean similarity baseline for deep LCAs, indicating that deeper regions of the tree correspond to semantically tighter groupings. This supports the claim that the learned tree induces a coherent hierarchy with increasing semantic refinement toward deeper levels.

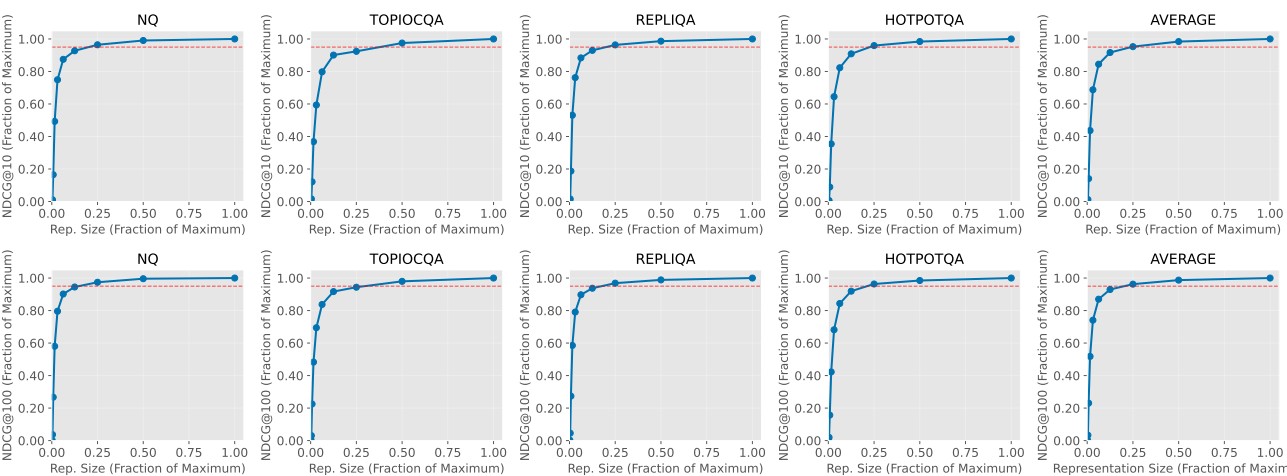

*Figure 13.* NDCG@10 (above) and NDCG@100 (below) performance vs. representation size trade-off for RETREEVER-STOCHASTIC. All values are normalized as fractions of their respective maximums. Dashed red lines show 95% performance thresholds. The average across datasets (rightmost) shows that 95% of peak performance is maintained even at 1/4 maximum representation size (i.e. dimension size 256) for NDCG@10 and at 1/8 maximum representation size (i.e. dimension size 128) for NDCG@100.

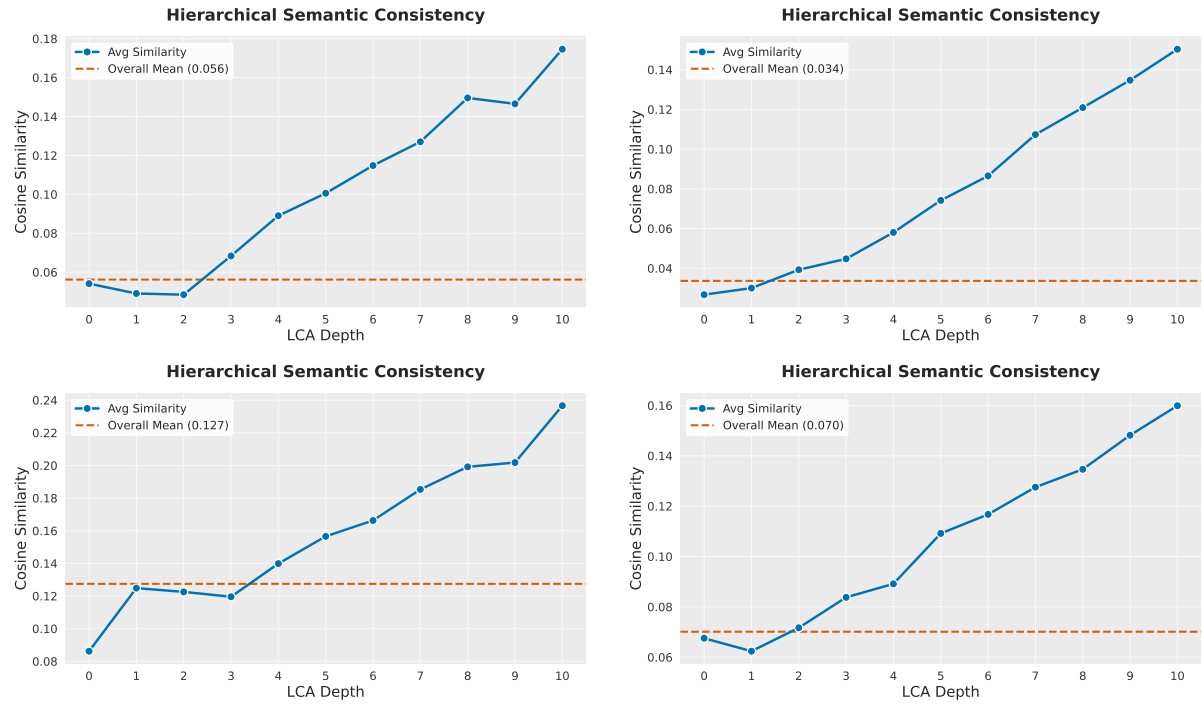

*Figure 14.* Cosine similarity vs LCA depth for text pairs

**Class-Node Alignment Across Tree Levels (ImageNet-1K).** To assess whether the learned tree induces semantically meaningful partitions on ImageNet-1K, we measure how well node assignments at each depth align with ground-truth class labels. For each tree level $\ell \in \{1, \ldots, D\}$, we extract the model's assignment scores over the $2^{\ell}$ nodes at that level (a contiguous slice of the full node-score vector), convert the soft assignments into a hard node index via $\arg\max$, and compute the *Normalized Mutual Information* (NMI) between the resulting node assignments and the ImageNet class labels. NMI is an information-theoretic measure of agreement between two clusterings, normalized to lie in $[0, 1]$, where $0$ indicates independence and $1$ indicates perfect correspondence. An increasing NMI with depth indicates that deeper levels of the tree progressively refine the partition of the dataset into more class-consistent groups, as expected from a coarse-to-fine

hierarchy. In our experiments, NMI rises consistently with tree depth and approaches $\approx 0.8$ at the deepest level, suggesting that the learned hierarchy strongly separates ImageNet classes into distinct regions of the tree rather than producing arbitrary splits. Figure 15 This supports the claim that ReTreever's routing structure captures meaningful semantic organization that becomes more discriminative as depth increases.

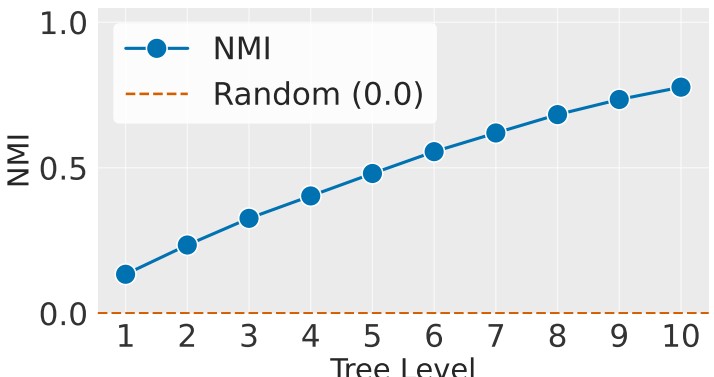

*Figure 15.* NMI between tree node assignments and imagenet1K classes increase with depth. Baseline is no correlation between the two having an NMI of 0.0

## I. Scaling to Web-Scale: the Reddit Title–Body Corpus

To stress-test RETREEVER beyond the million-document regime, we run an additional experiment on the `sentence-transformers/reddit-title-body` corpus.[6] This dataset contains **124,534,727 contexts**, making it roughly $300\times$ larger than NQ and $1700\times$ larger than TopiOCQA. Each query is the title of a Reddit submission, and the corresponding gold context is the body of that submission. We evaluate on a held-out set of 500 query–context pairs.

**Setup.** We train RETREEVER on top of the same `distilbert-base-uncased` encoder used in the main paper. We consider three tree depths: $D{=}10$ with 1024 leaves, $D{=}11$ with 2048 leaves, and $D{=}12$ with 4096 leaves. For each depth, we train both the constant-depth and stochastic-depth variants. All models are trained for 1.85M steps, corresponding to approximately one pass over the corpus, using the same contrastive objective as in the main experiments.

This setting is useful not only for measuring retrieval quality, but also for testing the practical indexing behavior of RETREEVER. A learned tree gives two independent control knobs. First, the tree depth determines the number of leaf partitions used for multi-index retrieval. Second, the level from which we take node-assignment representations determines the dimensionality of the vectors stored inside each leaf index. Thus, for a fixed trained tree, we can trade off index size, build time, search cost, and retrieval quality by choosing both the number of partitions and the representation size.

**Indexing at this scale.** At 124.5M contexts, standard flat or global approximate indexes become expensive to construct. In our experiments, ScaNN on the encoder representations exceeded 1 TB of RAM during construction and failed with an out-of-memory error under the available memory budget. In contrast, RETREEVER's multi-index construction completed successfully under the same budget. Excluding representation preprocessing and loading, building the 1024 leaf-level indices took 14 minutes when indexing 1024-dimensional tree representations, and only 2 minutes when indexing 256-dimensional tree representations. This is because RETREEVER first partitions the corpus using the learned hierarchy, after which retrieval reduces to exact $\ell_1$ search within the selected leaf partitions.

**Tree utilization at scale.** Figure 16 shows that the learned tree remains well utilized even at this much larger scale. With $D{=}10$, the stochastic variant partitions 124.5M documents across 1024 leaves, giving an average of 121,616 documents per leaf. The median leaf size is 57,006, the Gini coefficient is 0.61, and no leaves are empty. The distribution is naturally long-tailed, as expected for a heterogeneous web-scale corpus, but all leaves receive a substantial number of documents. This indicates that the learned hierarchy does not collapse to a small number of active regions, and instead produces a broad

---

[6]https://huggingface.co/datasets/sentence-transformers/reddit-title-body

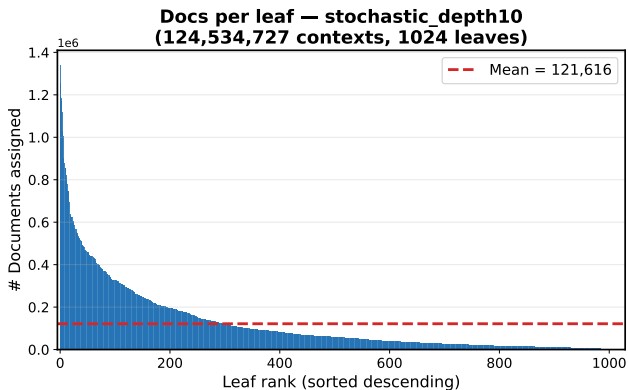
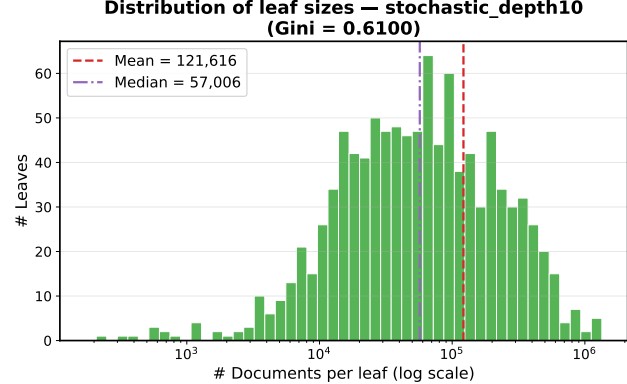

*Figure 16.* **Leaf utilization on Reddit Title–Body at** $D{=}10$**.** Left: number of documents per leaf, sorted in descending order. Right: distribution of leaf sizes on a log scale. The learned tree uses all 1024 leaves, with no empty leaves and Gini coefficient 0.61.

partition of the corpus suitable for multi-index retrieval.

**Multi-index retrieval trade-off.**    Figure 17 shows the quality–cost trade-off obtained by varying the fraction of documents accessed during multi-index retrieval. In this experiment, we use the depth-10 tree to partition the corpus into 1024 leaf indices, and vary the representation size used inside the indices. Larger representations provide higher retrieval quality, while smaller representations are cheaper to index and search. The 1024-dimensional variant reaches 14% nDCG@10 while accessing only 20% of the corpus, and lower-dimensional variants provide cheaper alternatives with a predictable reduction in quality. This illustrates the main practical benefit of the learned hierarchy at web scale: retrieval can be restricted to relevant tree partitions, and the cost within those partitions can be further controlled by choosing a coarser or finer representation level.

**Coarse-to-fine representations.**    Figure 18 evaluates the representation quality obtained at different tree levels. The same pattern observed on the smaller datasets also appears here: the stochastic-depth variant produces much stronger intermediate representations than the constant-depth model, and reaches high nDCG@10 at substantially smaller representation sizes than the encoder baseline. This confirms that stochastic depth training retains its intended behavior at web scale, producing useful coarse representations rather than only improving the finest leaf-level representation.

**Takeaway.**    The Reddit experiment shows that RETREEVER scales to a corpus with more than $10^8$ documents while preserving the same qualitative behavior observed in the main experiments. The learned tree uses all leaves, supports large but well-distributed leaf partitions, and enables efficient multi-index retrieval without requiring a global ANN index over the full corpus. Just as importantly, the hierarchy provides flexibility: we can choose the number of partitions through the tree depth, and choose the cost–quality point within each partition through the representation level. These properties make the learned tree especially useful in large-corpus settings, where indexing cost, memory footprint, and the number of searched documents become central practical constraints.

## J. Comparison of Different Propagation Methods

**Product Propagation.**    As introduced in Section 3.1, RETREEVER uses *product propagation* to couple node-wise routing scores through the geometry of the tree. The split function at each branching node produces a local routing logit, and applying a sigmoid yields a local branching probability. However, these local decisions must be consistent with the fact that reaching a node requires taking all routing decisions along the unique path from the root. Product propagation enforces this constraint by defining the probability of reaching node $t$ as the product of the relevant routing probabilities along its root-to-$t$ path. Concretely, given a node $t$ and its ancestors $\mathcal{A}_t$ (the nodes along the path from the root to $t$), the probability of an input $\mathbf{x}_i$ reaching $t$ is

$$T(\mathbf{x}_i)_t \;=\; \mathbf{z}_{i,t} \prod_{a \in \mathcal{A}_t} \mathbf{z}_{i,a}, \tag{7}$$

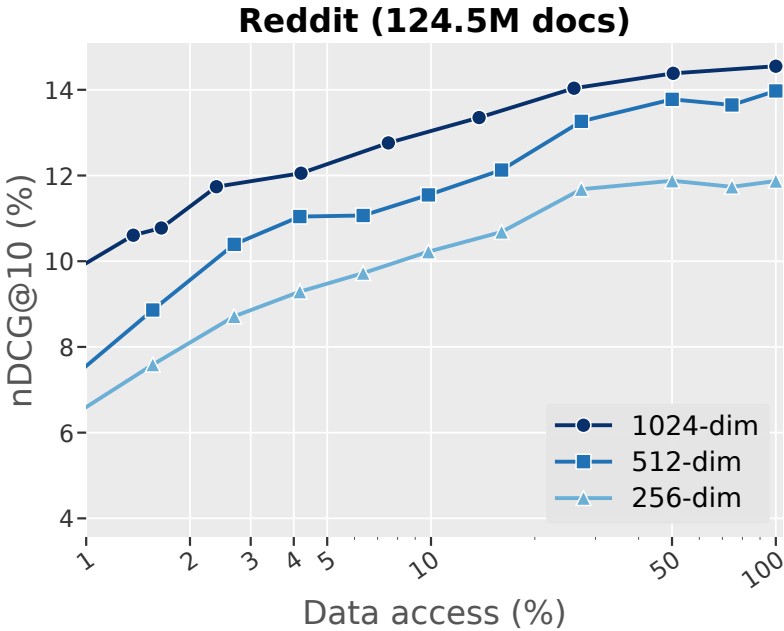

*Figure 17.* **RETREEVER on Reddit Title–Body with 124.5M documents.** nDCG@10 as a function of the fraction of documents accessed, shown for different representation sizes. Larger representations recover quality more quickly as the data budget increases; the 1024-dimensional variant reaches 14% nDCG@10 at only 20% data access.

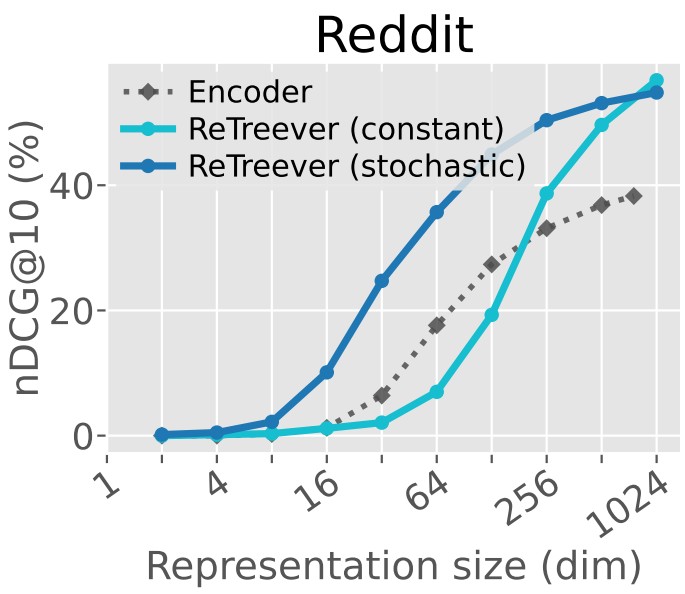

*Figure 18.* **Coarse-to-fine retrieval on Reddit Title–Body.** nDCG@10 as a function of representation size. The stochastic variant reaches high retrieval quality at substantially smaller representation sizes than the encoder baseline, showing that the coarse-to-fine behavior of RETREEVER transfers to the 124.5M-document setting.

where $\mathbf{z}_{i,u} \in [0,1]$ denotes the (sigmoid) routing probability produced at node $u$ for input $i$. This propagation rule yields a globally consistent distribution over nodes, and in particular a leaf-assignment distribution obtained by restricting $T(\mathbf{x}_i)$ to the leaf set.

**Path-Consistency Propagation (Relaxed Propagation).** While product propagation enforces a strict multiplicative constraint, we found that coarse representations can benefit from a more relaxed coupling of node probabilities. We therefore consider a relaxed alternative in which node scores are no longer treated as independent. Instead, for each node $t$, we rescore it using a learned function of the raw split logits along its path to the root. Let $r_{i,u} \in \mathbb{R}$ denote the raw split logit produced at node $u$ for input $i$. We define a path-conditioned rescoring function $f_\phi$ (implemented as a small MLP) that maps the sequence of logits on the root-to-$t$ path to a new node logit:

$$\tilde{r}_{i,t} \; = \; f_\phi\big(\{r_{i,u}\}_{u \in \mathcal{A}_t \cup \{t\}}\big), \qquad p_{i,t} \; = \; \sigma(\tilde{r}_{i,t}), \tag{8}$$

where $p_{i,t} \in [0,1]$ is now interpreted directly as the probability of *reaching* node $t$ (rather than a left/right branching probability at $t$). This introduces explicit dependence between a node and its ancestors through $f_\phi$, but does not by itself enforce tree-consistent paths; we therefore add a regularizer that encourages path consistency.

**Path-Consistency Regularization.** Given node-reaching probabilities $\{p_{i,t}\}_{t \in \mathcal{T}}$ for input $i$, the regularizer encourages high probability mass along the path to whichever node at a target depth is most likely. Let $\mathcal{T}_d$ be the set of nodes at depth $d$. We form a soft selection over nodes at depth $d$ using a temperature $\tau > 0$:

$$w_{i,t} \; = \; \frac{\exp(p_{i,t}/\tau)}{\sum_{u \in \mathcal{T}_d} \exp(p_{i,u}/\tau)}, \qquad t \in \mathcal{T}_d. \tag{9}$$

For each candidate node $t \in \mathcal{T}_d$, we score the log-probability of its entire root-to-$t$ path, and take a weighted average under $w_{i,t}$:

$$\mathcal{L}_{\mathrm{pc}}^{(d)} \; = \; -\frac{1}{B} \sum_{i=1}^{B} \sum_{t \in \mathcal{T}_d} w_{i,t} \sum_{u \in \mathcal{A}_t \cup \{t\}} \log\big(p_{i,u} + \varepsilon\big), \tag{10}$$

where $B$ is the batch size and $\varepsilon$ is a small constant for numerical stability. As $\tau \to 0$, the weights concentrate on the highest-probability node at depth $d$, recovering a hard "best-path" penalty; for larger $\tau$, the loss encourages consistency across multiple plausible paths.

**Final Objective.** We add the path-consistency term to the main contrastive objective in Equation (1). Since we compute node probabilities for both queries and contexts, we apply the regularizer to both:

$$\mathcal{L} \; = \; \mathcal{L}_{\mathrm{InfoNCE}} \; + \; \lambda\Big(\mathcal{L}_{\mathrm{pc}}(\mathrm{queries}) + \mathcal{L}_{\mathrm{pc}}(\mathrm{contexts})\Big), \tag{11}$$

where $\lambda$ controls the strength of the regularization. We compare strict product propagation against the relaxed MLP propagation with path-consistency regularization in Table 10, and we see the level wise difference in Figure 19.

We observe that while both propagation schemes achieve similar performance at the finest level, the relaxed propagation yields stronger coarse representations, most notably on RepLiQA. In particular, on RepLiQA it also outperforms the MRL baseline in Appendix B at coarse granularities. For simplicity, however, we use product propagation in all other experiments.

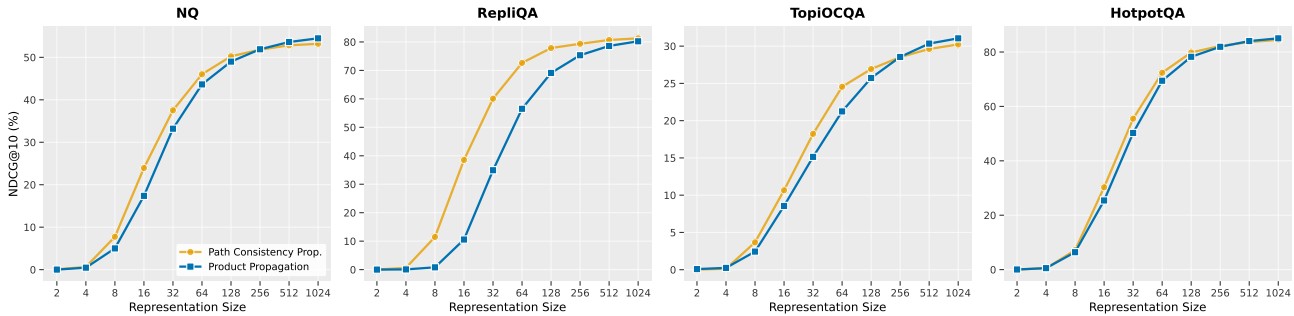

*Figure 19.* Level wise retrieval performance for different propagation methods

*Table 10.* Effect of different propagation mechanisms on text retrieval performance (NDCG@10). Best results per dataset within each block are in bold.

| MODEL | NQ | HOTPOTQA | TOPIOCQA | REPLIQA |
|---|---|---|---|---|
| **PRODUCT PROPAGATION** | | | | |
| RETREEVER - STOCHASTIC | 0.5448 | 0.8504 | 0.3100 | 0.8023 |
| RETREEVER | **0.5458** | **0.8571** | 0.3178 | **0.8222** |
| **PATH CONSISTENCY PROPAGATION** | | | | |
| RETREEVER - STOCHASTIC | 0.5417 | 0.8379 | 0.3267 | 0.8097 |
| RETREEVER | 0.5374 | 0.8457 | **0.3429** | 0.8122 |

# K. Leaf Occupancy and Routing Entropy

For a binary ReTreever tree of depth $T$, the terminal level contains $2^T$ leaves. In our main experiments, we use $T = 10$, yielding 1024 leaves. The topology is balanced in the sense that all root-to-leaf paths have the same length and the finest-level representation has a fixed dimensionality. However, the allocation of documents to leaves is determined by the learned routing functions and is not constrained to be uniform. The resulting leaf occupancy therefore depends both on the structure of the dataset and on how the retrieval objective organizes the corpus.

To contextualize the learned tree occupancies, we first examine the distribution of documents in the frozen encoder space before training ReTreever. We run $k$-means with 1024 clusters on the frozen encoder representations and sort clusters by the number of assigned documents. As shown in Fig. 20, the resulting cluster sizes are highly non-uniform across all datasets. The largest clusters contain 774, 608, 334, and 214 documents for NQ, HotpotQA, RepLiQA, and TopiOCQA, respectively, while the corresponding mean cluster sizes are 72, 89, 69, and 30. This indicates that the underlying corpora already exhibit substantial variation in semantic density: some regions of the encoder space contain many more documents than others.

We observe a related long-tailed pattern in the learned ReTreever leaf assignments. For each document, we assign it to the leaf with maximum routing probability and report the sorted leaf occupancies in Fig. 21. The learned distributions are non-uniform, reflecting the non-uniform structure of the datasets, but the tree still uses a broad set of leaves. For instance, on NQ, the largest learned leaf contains 349 documents and no leaves are empty. Compared to the corresponding 1024-way $k$-means partition, the learned tree reduces the size of the largest occupied region while maintaining broad coverage over the leaf set. RepLiQA has 184 empty leaves and TopiOCQA has one empty leaf, but the majority of leaves remain active in both cases. Overall, these results suggest that ReTreever learns a leaf distribution that follows the natural skew of the data while still spreading documents across much of the available tree capacity.

We further quantify soft leaf utilization using the entropy of the routing distribution. For an input document $x$, ReTreever produces a probability distribution over leaves,

$$p_x(\ell), \qquad \ell \in \{1, \ldots, 2^T\}.$$

We compute the leaf-routing entropy in bits as

$$H(x) = -\sum_{\ell=1}^{2^T} p_x(\ell) \log_2 p_x(\ell).$$

For a depth-$T$ tree, the maximum possible entropy is $T$ bits, achieved by the uniform distribution over all $2^T$ leaves. In our setting, $T = 10$, so the maximum entropy is 10 bits. We report the mean entropy across documents,

$$\bar{H} = \frac{1}{N} \sum_{i=1}^{N} H(x_i).$$

The mean tree entropies are 8.40 bits for NQ, 8.74 bits for HotpotQA, 7.85 bits for RepLiQA, and 8.00 bits for TopiOCQA. These values indicate that the learned routing distributions retain substantial support over many leaves, rather than concentrating all probability mass on a single path. This soft-utilization view complements the hard occupancy plots: the hard

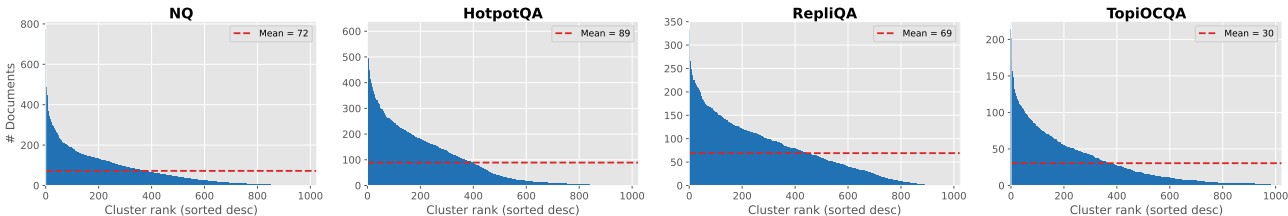

*Figure 20.* Document distribution across 1024 $k$-means clusters computed from frozen encoder representations. Clusters are sorted by occupancy in descending order. Dashed red lines indicate the mean number of documents per cluster. The distributions show that the frozen encoder space is already substantially non-uniform across datasets.

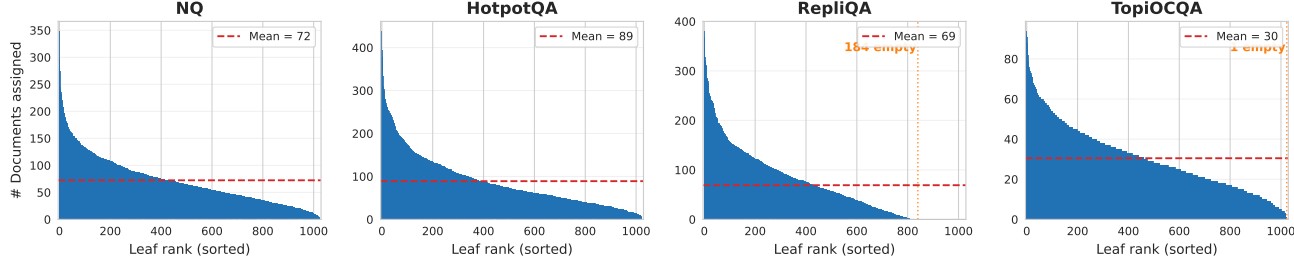

*Figure 21.* Document distribution across the 1024 leaves of the learned ReTreever tree. Each document is assigned to the leaf with maximum routing probability, and leaves are sorted by occupancy. Dashed red lines indicate the mean number of documents per leaf. The learned tree exhibits non-uniform occupancy consistent with dataset skew, while maintaining broad utilization of the leaf set.

assignments show how documents are distributed under argmax leaf assignment, while the entropy measures how broadly each document's routing probability is spread over the terminal level.

Together, these diagnostics show that ReTreever's fixed-depth tree produces a non-uniform but broadly utilized hierarchy. The non-uniformity is consistent with the skew already present in the datasets and frozen encoder space, while the learned routing objective shapes how this skew is expressed across the tree leaves.

## L. Method Scope and Extensions

### L.1. Relationship to Encoder Adaptation

RETREEVER operates on top of a frozen bi-encoder, and in this sense it can be viewed as a lightweight trainable module attached to a fixed representation backbone. However, the learned tree plays a broader role than a standard encoder adapter. It defines a retrieval-optimized hierarchy whose node assignments serve as coarse-to-fine representations, an inspectable organization of the corpus, and the partitioning structure used for multi-index retrieval. Thus, the added parameters do not only transform the encoder embedding space, but also induce the structure through which documents are organized and searched.

This perspective also motivates our choice of lightweight baselines on the same frozen encoder. Linear, MLP, and MRL adapters isolate whether improvements come primarily from adding trainable parameters on top of the encoder, or from the learned hierarchical structure itself. Fully fine-tuning the encoder, or applying parameter-efficient fine-tuning directly to the encoder backbone, is an important complementary regime and can provide a useful raw-performance ceiling. However, it changes the comparison setting: RETREEVER is designed to study what can be gained by learning a retrieval-optimized hierarchy while keeping the underlying encoder fixed.

### L.2. Hierarchical Representations and Search

We refer to RETREEVER as a hierarchical retriever because the tree shapes both the learned representations and the search procedure. Every representation is obtained by routing an input through the tree and combining node decisions via tree propagation, so the leaf-assignment vector depends on the full ancestor structure rather than on an unconstrained flat projection. Intermediate tree levels also provide explicit coarse representations, enabling retrieval at different granularities

by choosing the corresponding level of the hierarchy.

The hierarchy is also used directly during multi-index retrieval. Documents are assigned to leaf-level indices, and each query is routed through the same tree to select the most relevant leaves before local similarity search is performed. In this setting, flat vector comparison is only the final scoring step within the selected partitions; candidate selection itself is governed by the learned hierarchy. The same tree therefore provides a unified structure for representation learning, corpus organization, and efficient search.

### L.3. Adapting the Tree to Evolving Document Corpora

RETREEVER uses a fixed binary tree topology once training is complete. For a depth-$T$ tree, all root-to-leaf paths have length $T$, and the terminal level contains $2^T$ leaves. This fixed topology makes the representation size and inference procedure predictable, but it also means that the tree capacity is allocated uniformly in structure rather than adaptively in response to local corpus density. Since document routing is learned from data, the resulting leaf occupancies can still be highly non-uniform: dense semantic regions may assign many documents to the same leaf, while sparse regions may use only a few leaves. Such overloaded leaves are most relevant for multi-index retrieval, where search cost depends on how many documents are stored in the selected leaves. They may also indicate regions where the corpus would benefit from finer semantic subdivision. Natural extensions include adaptive-depth or variable-width trees, local refinement of overloaded leaves, and jointly learning the topology together with the routing functions.

RETREEVER also supports insertion of new documents without retraining. A new document is encoded with the frozen encoder, routed through the learned tree, and inserted into the corresponding index or leaf-level indices. This procedure is suitable when incoming documents are drawn from a distribution similar to the training corpus, since the learned hierarchy remains a reasonable semantic organization of the data. Under substantial corpus drift, however, the tree remains usable but may no longer provide the most effective organization: new documents may concentrate in a small number of leaves, or previously sparse regions of the tree may become more important. This behavior is analogous to distribution shift in dense retrieval systems, where a retriever trained on one corpus may generalize imperfectly to a changed or expanded corpus. Promising directions for dynamic settings include periodic tree refreshes, lightweight distribution-specific adapters, local updates to selected subtrees, and topology modifications that selectively refine overloaded or shifted regions while preserving the rest of the learned hierarchy.

## M. Interactive Tree Visualization Tool

To facilitate the inspection and analysis of learned hierarchical structures in RETREEVER, we developed an interactive web-based visualization tool that renders the complete binary tree structure using Plotly.js, with automatic layout positioning based on node depth and rank. We make this tool available here: https://github.com/ServiceNow/retreever

**Construction**  In the trees representing text datasets, upon selecting any node, the tool displays comprehensive metadata including the node identifier and tree level, an automatically generated semantic label, a summary description of the assigned document cluster, and dominant keyword themes. We use the `Qwen/Qwen2.5-72B-Instruct` model to generate this semantic metadata. For leaf nodes, we prompt the model to extract the most specific, fine-grained topics shared by documents assigned to the node. For internal nodes, we prompt the model to (i) identify salient keywords from a random subset of highly ranked documents in the node's cluster, and (ii) jointly analyze the child nodes' summaries and keywords to infer the broader concept represented by the parent. In addition, we ask the model to characterize how the left and right subtrees differ by producing child-specific keywords that are indicative of one child (left or right) but not the other, highlighting both shared domains and discriminative themes.

**Features**  The tool supports interactive path highlighting: selecting a node highlights the full root-to-node path and shows each ancestor's label and key keywords inline, making semantic refinement easy to track (see Figure 22). Navigation is streamlined via clickable ancestor breadcrumbs and left/right branch panels, each annotated with keywords for quick semantic orientation. A full-text search runs over node labels, summaries, and keywords, returning results grouped by tree level while simultaneously highlighting matches in the tree. The search box includes autocomplete over all node labels for direct jumps to relevant subtrees. For each node, we display top dominant themes and branch-distinctive keywords that summarize how the left and right children diverge (see Figure 23).

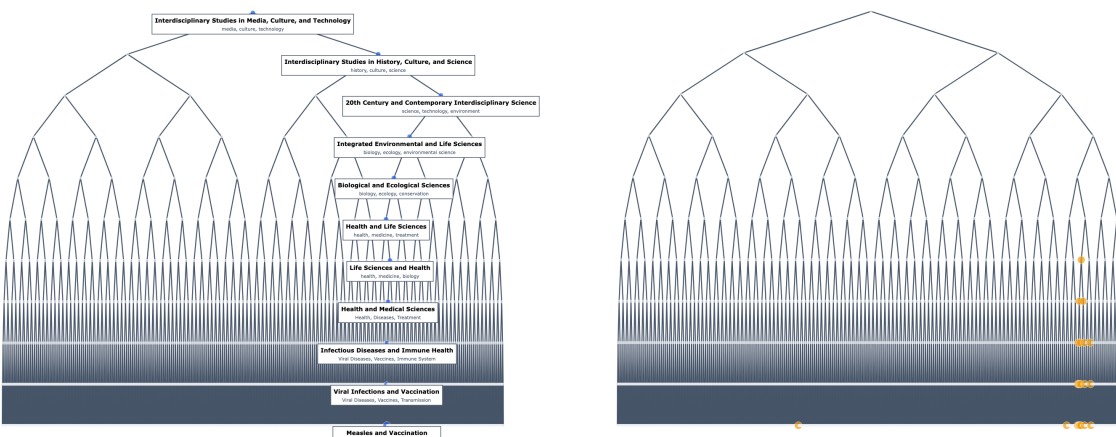

*Figure 22.* Interactive visualization tool features. *Left:* Node selection highlights the root-to-node path with inline metadata annotations showing labels and keywords for each ancestor. *Right:* Text search for "physics" highlights matching nodes in the tree.

**Case Studies: Hierarchical Organization**    The tree visualization provides an intuitive view of semantic refinement across levels, making it easy to understand how RETREEVER organizes the corpus into progressively finer concepts. For example, in Figure 23, the left panel highlights a clear refinement path (Entertainment → Contemporary Films → Notable Directors → James Cameron), while the right panel reveals a meaningful split of the director's filmography into the *Terminator* franchise versus other major works (e.g., *Avatar*, *Titanic*). Additional examples in Figure 24 further illustrate this refinement behavior: one branch narrows from Science → Biology → Health → Medicine to Infectious Diseases and ultimately HIV, while another specializes from Media → Entertainment → Television to British Television and finally to *Doctor Who* (Season 11). These paths show that the tree captures both broad topical structure and highly specific subtopics at deeper levels.

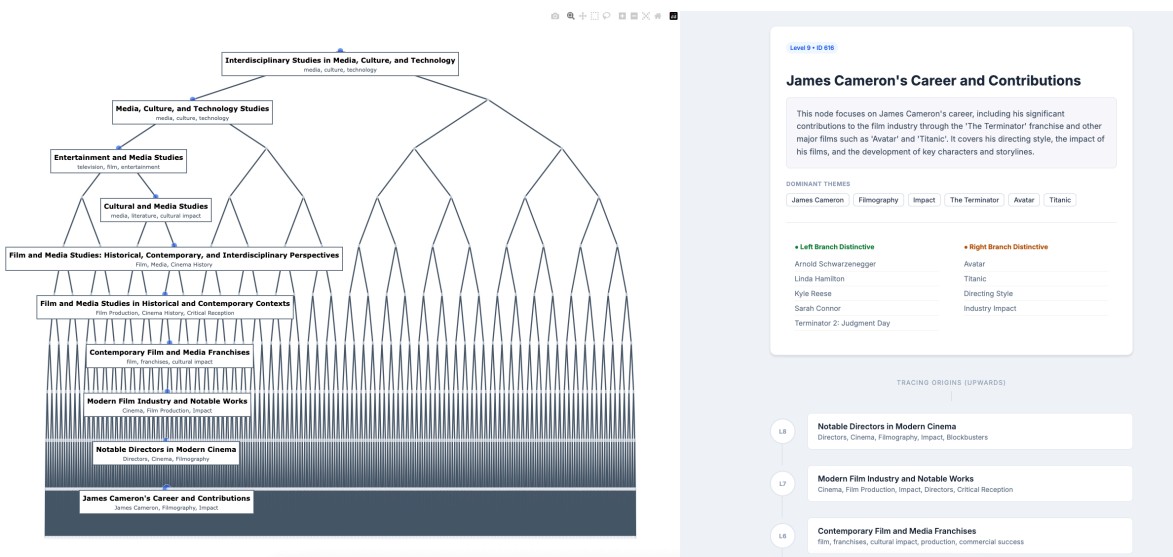

*Figure 23.* Visualization interface showing the interactive tree (left) and node metadata panel (right). The tree reveals a clear semantic refinement path (Entertainment → Contemporary Films → Notable Directors → James Cameron), while the metadata highlights the split into the *Terminator* franchise (left subtree) versus other works such as *Avatar* and *Titanic* (right subtree).

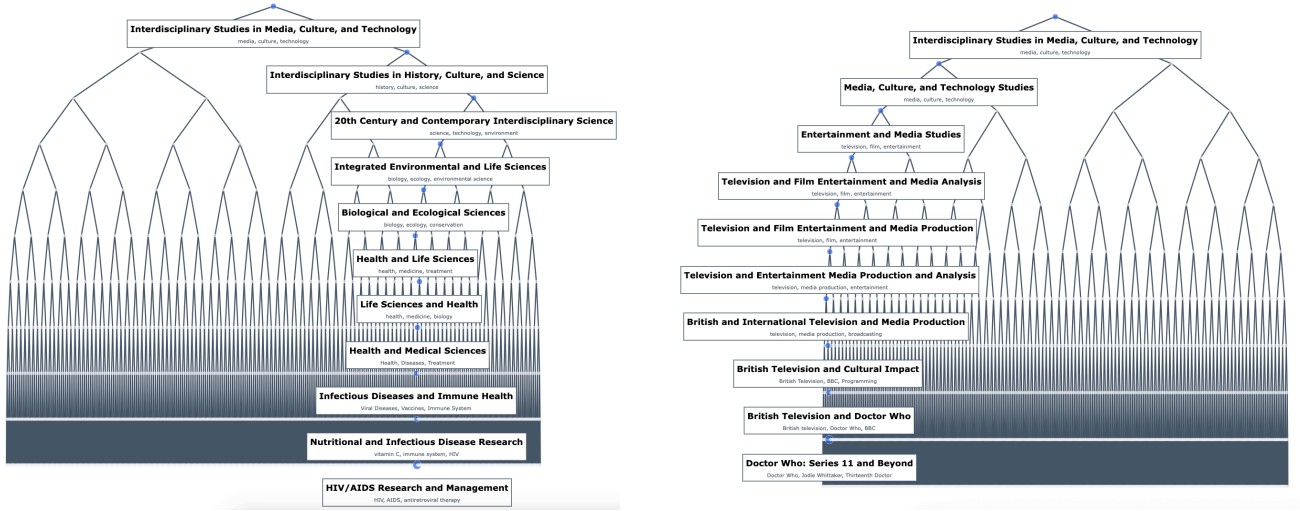

*Figure 24.* Examples of hierarchical document organization. *Left*: Science → Biology → Health → Medicine → Infectious Diseases → HIV. *Right*: Media → Entertainment → Television → British Television → Doctor Who → Doctor Who (Season 11).

**ImageNet1K-Based Trees**   We next discuss RETREEVER trees trained on ImageNet1K for image-to-image retrieval, where the goal is to retrieve other images from the same class as a given query image. In this setting, the learned hierarchy primarily reflects visual similarity. For instance, Figure 25 shows a node that separates modern agricultural machinery (e.g., combine harvesters, plows, snowplows) in the left subtree from more primitive transport and field-work scenes in the right

subtree (e.g., horse carts). The appearance of bison/oxen in the right subtree suggests that the split is driven by shared visual cues such as animals pulling carts in open fields, which can resemble harvester imagery at a coarse level while remaining semantically distinct, yielding a clean left/right partition.

Similarly, in Figure 26, the selected node splits into two visually coherent subtrees. The left subtree groups coral leaves, honeycomb, strainers, and brain coral, which share strong lattice-like, porous, or repetitive textures. The right subtree contains birds such as the red-backed sandpiper and dowitcher; as waders, they are frequently photographed in shoreline scenes whose backgrounds (water, wet sand, ripples) can overlap visually with reef-like textures. At this level, the node captures a broad notion of "near-water / high-frequency texture" similarity, while its children separate these into texture-dominated objects (left) versus animal-centric shoreline scenes (right). This behavior is a direct consequence of the image-to-image retrieval objective, which prioritizes visual similarity over semantic relatedness.

Finally, this kind of visually driven routing can also yield groupings that sound nonsensical textually, but are coherent in pixel-space. For example, in Figure 27, a node clusters ski masks with rock beauty fish. While semantically unrelated, both categories frequently exhibit high-contrast, saturated color patterns (notably yellow, black, and red), and the node appears to capture this shared visual motif; the children then separate the two classes. Similarly, Figure 28 shows a node grouping elephants with cocktail shakers, which becomes intuitive on inspection: the elongated, curved silhouette of a shaker can resemble an elephant's trunk. The subsequent split refines this coarse visual similarity into distinct subclusters, with elephants in the left subtree and cocktail shakers in the right.

We hope readers find our visualization tool useful and enjoyable to explore, and that it highlights how trees learned by RETREEVER can be readily converted into artifacts for direct visual inspection and analysis.

## N. Hyperparameters

This appendix details the hyperparameters used for training RETREEVER models across different modalities and datasets.

### N.1. Common Training Hyperparameters

Table 11 lists hyperparameters shared across all RETREEVER variants.

*Table 11.* Common training hyperparameters for all ReTreever models.

| Hyperparameter | Value |
| --- | --- |
| Optimizer | AdamW |
| Learning rate | 4e-4 |
| Weight decay | 0.01 |
| Warmup steps | 10,000 |
| LR Scheduler | Linear |
| Total training steps | 200,000 |
| Batch size | 64 |
| Gradient clipping | 1.0 |
| Tree depth ($D$) | 10 |
| Maximum leaves | 1,024 |
| Loss function | InfoNCE Equation (1) |
| Similarity metric | Negative TVD |

### N.2. Architecture Hyperparameters

Table 12 describes the architecture-specific hyperparameters for the tree structure and split functions.

### N.3. Encoder-Specific Hyperparameters

Table 13 details encoder configurations across all modalities. For text retrieval we use DistilBERT (trained on MS MARCO) as our primary encoder and BGE-large as a secondary comparison. Image retrieval experiments use three vision encoders on ImageNet-1K, and audio retrieval uses AST on VoxCeleb2.

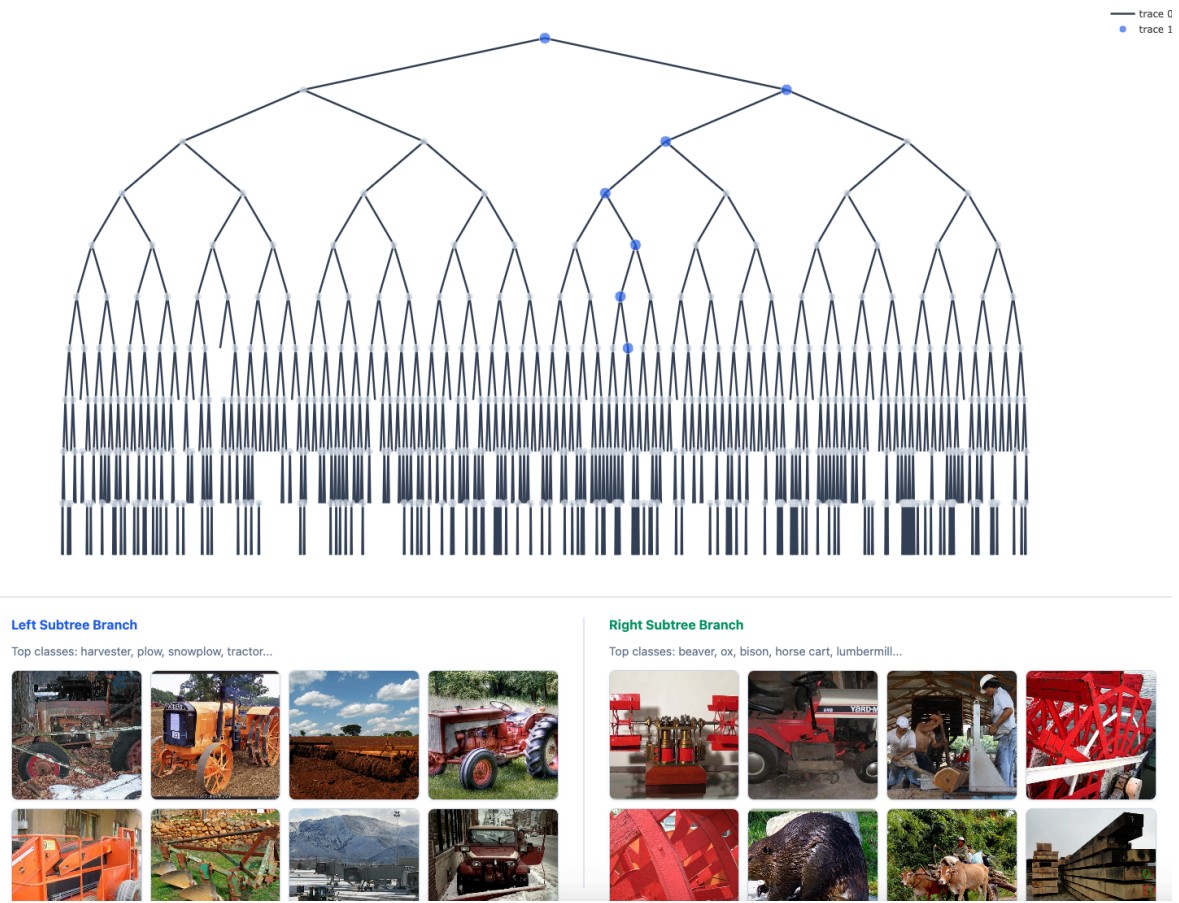

*Figure 25.* Example of a learned ImageNet1K hierarchy for image-to-image retrieval. The selected node partitions visually similar images into two coherent subclusters: modern agricultural machinery (left; e.g., harvesters, plows, snowplows) versus more primitive field-transport scenes (right; e.g., horse carts, oxen/bison pulling carts). This illustrates how the tree refines coarse visual similarity into finer-grained groupings.

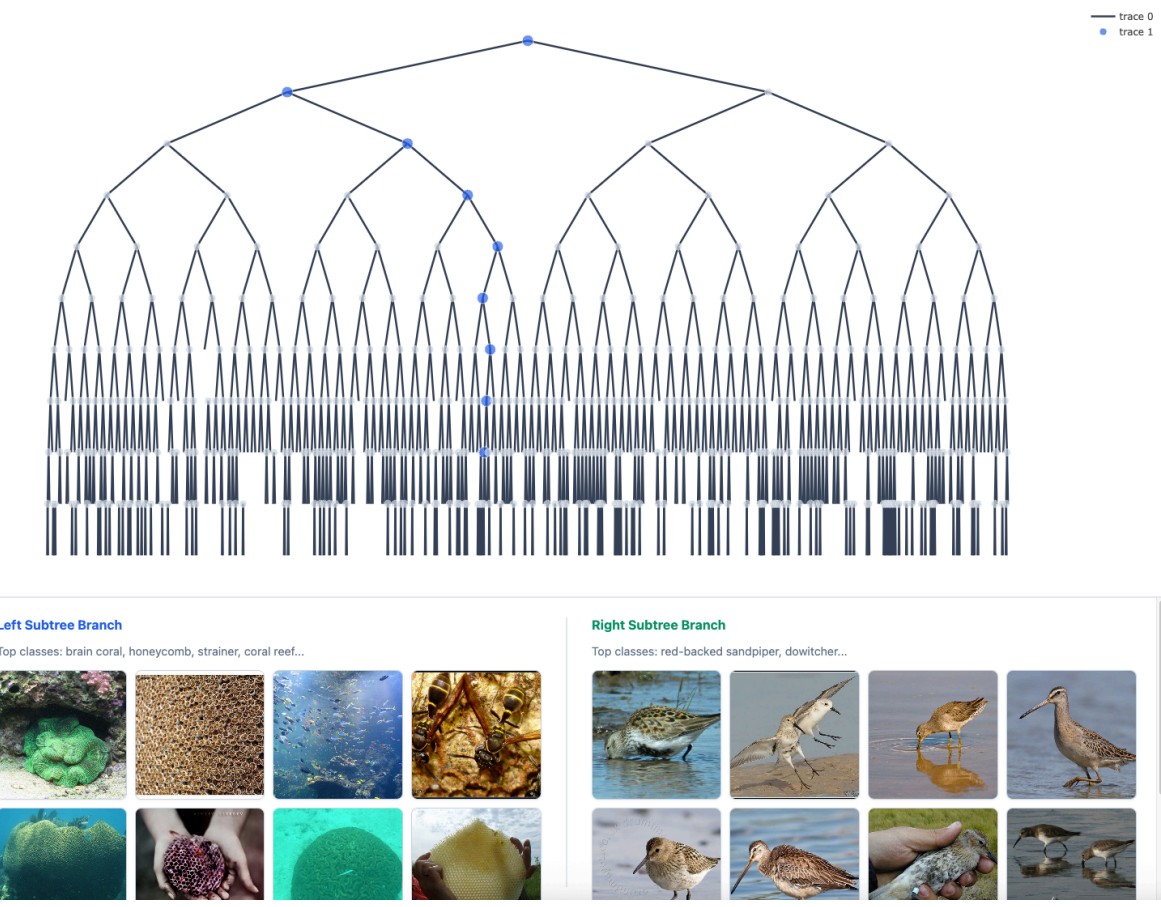

*Figure 26.* Example of visually driven hierarchy in an ImageNet1K ReTreever tree. The selected node groups images with similar high-frequency, porous or lattice-like textures. Its left subtree contains texture-dominated objects (coral leaves, honeycomb, strainers, brain coral), while its right subtree contains shoreline bird images (e.g., red-backed sandpiper, dowitcher). The split illustrates how the tree first captures broad visual similarity and then refines it into more specific visual concepts, consistent with an image-to-image retrieval objective.

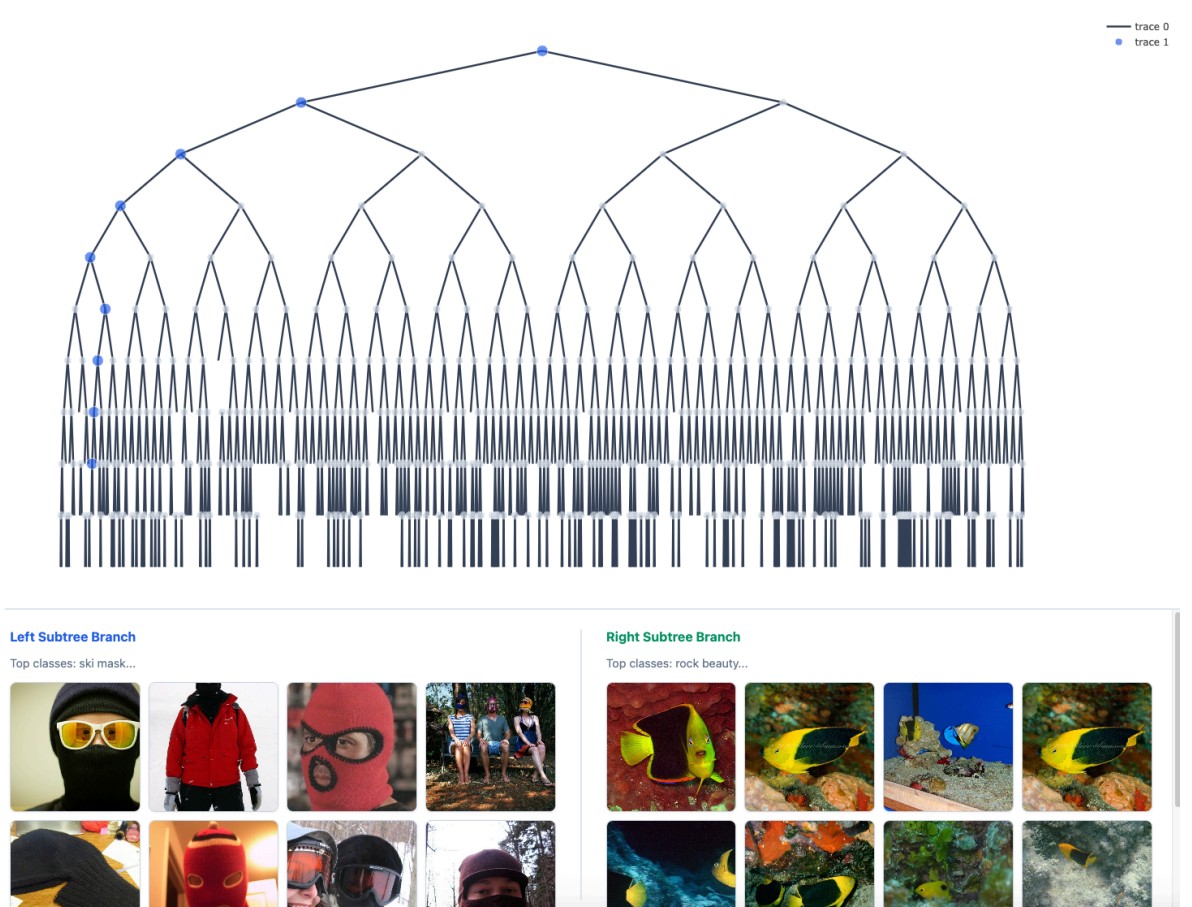

*Figure 27.* A visually coherent but semantically unexpected node: ski masks and rock beauty fish are grouped due to shared high-contrast, saturated color patterns (e.g., yellow/black/red), with the children splitting the two categories into separate subclusters.

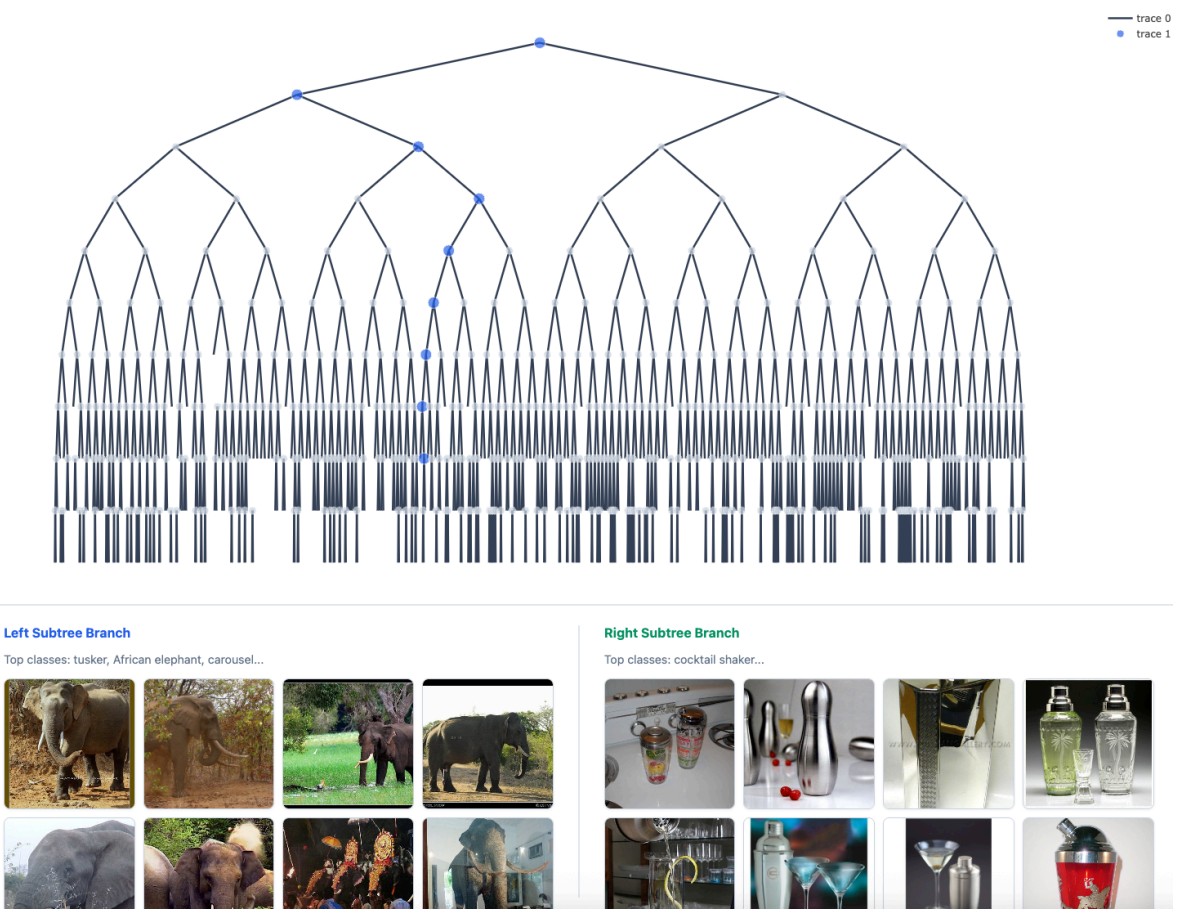

*Figure 28.* A shape-driven grouping: elephants and cocktail shakers co-occur at an internal node, likely reflecting a shared elongated curved silhouette (trunk-like geometry), before the children refine the cluster into elephants (left) versus cocktail shakers (right).

*Table 12.* ReTreever architecture hyperparameters.

| Hyperparameter | Value |
|---|---|
| *Split Function* | |
| Split function type | Cross-attention |
| Number of attention heads | 16 |
| Attention head dimension ($d_k$) | 64 |
| Total split function dimension | 1024 |
| Embeddings per level ($n_l$) | 8 |
| Level embedding dimension | 1,024 |
| *Tree Propagation* | |
| Propagation method | product-propagation |
| Aggregation function | Per-node linear projection + mean |
| *Depth Scheduling* | |
| Constant depth | Level 10 only |
| Stochastic depth | Levels 1–10, $p(\ell) \propto \ell^2$ |
| Trainable parameters added by RETREEVER | 3.7M |

*Table 13.* Encoder hyperparameters across modalities.

| Hyperparameter | Text | | Image | | | Audio |
| | DistilBERT | BGE | DinoV2 | ResNet50 | CLIP-ViT-L | AST |
|---|---|---|---|---|---|---|
| Base encoder | distilbert-base (MS MARCO) | bge-large-en v1.5 | dinov2-large | resnet50 | clip-vit-large patch14 | ast-audioset |
| Number of parameters | 66M | 335M | 407M | 26M | 304M | 86M |
| Encoder dimension | 768 | 1,024 | 1,024 | 2,048 | 768 | 768 |
| Max sequence length | 512 tokens | 512 tokens | – | – | – | – |
| Image resolution | – | – | 224×224 | 224×224 | 224×224 | – |
| Audio sampling rate | – | – | – | – | – | 16 kHz |
| Audio duration | – | – | – | – | – | 10 sec |

## N.4. Hardware

All models were trained on NVIDIA H100 80GB GPUs.

