# OpenReview forum: "Hierarchical Retrieval at Scale: Bridging Interpretability and Efficiency"
_ICML.cc/2026/Conference — ICML 2026 regular_

### Official Review · Reviewer_V8Q7 · 2026-03-09

**Soundness:** 2
**Presentation:** 3
**Significance:** 3
**Originality:** 2
**Overall Recommendation:** 3
**Confidence:** 2

**Summary:**

This paper proposes ReTreever, a hierarchical retrieval framework. ReTreever introduces a differentiable binary tree-based solution. The model learns a tree structure on top of frozen encoders (e.g., BERT, DINOv2) and optimizes the routing function via contrastive learning, so that queries and positive document samples have similar routing paths in the tree. ReTreever generates coarse-to-fine multi-granularity representations, enabling not only efficient retrieval but also interpretable semantic grouping through the tree structure.

**Main Contributions:**
1. Proposal of the ReTreever Framework: It designs a differentiable tree structure for end-to-end optimization of retrieval tasks. This structure constructs hierarchical indices without requiring LLMs, significantly improving the efficiency of hierarchical retrieval.
2. Balance between Performance and Efficiency: It is validated on text, image, and speech multimodal datasets, achieving the best trade-off between latency and retrieval accuracy, significantly outperforming LLM-based hierarchical retrieval methods (e.g., Raptor) and simple hierarchical clustering methods.
3. The paper demonstrates that ReTreever, while only optimizing for retrieval objectives, naturally learns semantically coherent document groupings. This makes the retrieval process more interpretable and supports flexible cost control (through representations at different levels).

**Compliance With Llm Reviewing Policy:**

Affirmed.

**Final Justification:**

The authors have addressed most of my concerns, so I’m changing the score to 3.

**Key Questions For Authors:**

1. The current training pipeline is offline, constructing the tree based on a static dataset. In real-world retrieval scenarios, data is often dynamically added. If a large number of new documents are introduced, does ReTreever require retraining the entire tree? If only local adjustments are needed, how can the mismatch between the distribution of new documents and the existing tree structure be handled?
2. During inference, when ReTreever uses "multi-index search," it retrieves only the top-k leaf nodes. In tasks requiring extremely high recall (e.g., >99%), could this tree-based pruning strategy become a bottleneck? Is there a risk that routing errors might cause critical documents to be pruned too early?
3. The paper fixes the tree depth to 10. For massive corpora e.g., billions of items, a fixed-depth binary tree might overload leaf nodes with too many documents, thereby reducing retrieval accuracy. Does ReTreever support unbalanced trees or variable-depth extensions?

**Limitations:**

Yes.

**Strengths And Weaknesses:**

**Strengths**
1. The paper is logically structured and well-presented. Figure 1 clearly illustrates the training and inference processes, and the formulation of key components (e.g., loss functions and routing probabilities) is explicit.
2. This research addresses a critical pain point: the high computational cost and black-box nature of dense retrieval. With the growing popularity of RAG, large-scale retrieval with low cost and high transparency is hot topic for both industry and academia. ReTreever offers an LLM-free and interpretable solution, which holds good practical value.

**Weaknesses**
1. Although Raptor is used as a baseline, Raptor itself is designed for long-document summarization and question answering. Directly using it as a baseline for short-text retrieval may introduce fairness biases.
2. Additionally, the quantitative analysis of "interpretability" mainly relies on auxiliary tools and indirect metrics (e.g., node purity), lacking direct human evaluation or quantification of the downstream task benefits (e.g., speed of error diagnosis) brought by "transparency."
3. The core technical components (e.g., differentiable trees, contrastive learning) are drawn from existing work. The paper's novelty lies more in the successful integration of these techniques and their specific optimization for the retrieval domain.

---

> ### Author Rebuttal · Authors · 2026-03-31
>
> We thank the reviewer for highlighting the paper’s clear presentation, the practical importance of efficient and inspectable retrieval, and ReTreever’s value as an LLM-free differentiable tree framework with strong latency-accuracy trade-offs across modalities.
>
> **W1**: While RAPTOR was introduced for long-document QA/summarization, its core mechanism is directly relevant here: it recursively clusters and summarizes embedded text chunks into a hierarchy. Our text datasets also operate over short retrieved passages, so we view RAPTOR as a reasonable LLM-based hierarchical baseline, while agreeing that it is not a perfect match and will clarify this more carefully in the revision.
>
> **W2**: We agree that “inspectable semantic structure” is more precise than a broad claim of full transparency. Our point is that the learned tree exposes where a miss occurs and which semantic split separated query and context. For example, we observed that a query about Muhammad Ali’s Olympic gold medal routed toward a sports/competition branch while the relevant biography-oriented context routes toward a biography/life-history branch, making the failure mode visible in a way flat dense embeddings do not. We will add appendix case studies with the query, ground-truth context, routing paths, and first split point responsible for the miss.
>
>
> **W3**: While ReTreever builds on ingredients that have appeared separately in prior work, we believe the contribution is more than their integration. The main contribution is a retrieval-specific formulation that combines properties uncommon in prior work: an LLM-free hierarchical retriever with coarse-to-fine representations, an inspectable semantic hierarchy, and tree-induced partitions for efficient search. Concretely, our instantiation introduces and combines (i) a cross-attention split function over token-level representations for routing, which is more expressive than standard linear splits, and (ii) a contrastive training objective that optimizes the tree directly for query-context retrieval rather than for classification, regression, or clustering as in previous differentiable tree learning literature. More broadly, the same learned tree simultaneously serves as the representation space, the search partition, and the inspectable semantic structure, making the overall formulation retrieval-specific and nontrivial.
>
> **Q1**: For new in-distribution documents, ReTreever does not require retraining: documents are encoded with the frozen encoder, routed through the learned tree, and inserted into the corresponding indices. The harder case is when enough new documents arrive that the corpus semantics themselves shift and the learned hierarchy should evolve. In that regime, the current tree remains usable, but it may no longer be the optimal semantic organization, since it was learned for the original document distribution. This is a broader dense-retrieval issue rather than something unique to ReTreever; e.g., BEIR notes that strong dense retrievers can generalize poorly under distribution shift [1]. Possible efficient extensions include local tree updates, lightweight shift-specific adapters, or selective topology refinement in overloaded regions; we will clarify that these are future-work directions beyond the current scope.
>
> **Q2**: For very high recall, multi-index search is a tunable trade-off rather than a hard bottleneck. As shown in Fig. 6 [0], which mirrors the multi-index retrieval setup, increasing k (the number of leaves searched per query) quickly brings the query closer to the positive document’s leaf: for NQ, RepLiQA, and TopiOCQA, the mean deepest LCA is already around 7-9/10 at k=10 (about 1% of leaves searched) and approaches 10/10 as k grows, while HotpotQA improves more gradually because its evidence often spans multiple concepts. The fraction of queries whose positive document leaf is searched rises from 54.0 / 17.6 / 63.7 / 41.8% at k=10 to 85.4 / 50.4 / 93.4 / 78.2% at k=100 for NQ / HotpotQA / RepLiQA / TopiOCQA. For near-exhaustive recall, one can search more leaves or use single-index retrieval, which does not prune leaves.
>
> **Q3**: Depth 10 is the current design choice, not the only topology the framework could support. Even so, fixed depth already scales much further than the main-paper benchmarks: in the rebuttal we successfully trained and indexed ReTreever on the 124.5M-context Reddit Title-Body corpus (Reviewer ttPY, W2/Q2, Fig. 4 [0]). With a fixed-depth tree, more documents per leaf mainly affects the efficiency of multi-index search rather than the quality of the learned routing itself, since retrieval can still use the full routing distribution. For still larger corpora, deeper trees, ANN search within leaves, or more adaptive topologies (e.g., splitting overloaded leaves, variable-depth growth in dense regions, or non-uniform branching) are natural extensions.
>
> [0] https://anonymous.4open.science/r/b-EDFB/e.pdf
>
> [1] Thakur et al., 2021, arXiv:2104.08663.

---

> > ### Author Rebuttal · Reviewer_V8Q7 · 2026-04-03
> >
> > The authors have addressed most of my concerns, so I’m changing the score to 3.

---

> > > ### Author Response · Authors · 2026-04-05
> > >
> > > Thank you for your thoughtful reconsideration and for updating your score. We appreciate that our rebuttal addressed your concerns, and we are grateful for the time and care you gave to re-evaluating the paper.

---

### Official Review · Reviewer_D6Se · 2026-03-11

**Soundness:** 2
**Presentation:** 2
**Significance:** 2
**Originality:** 2
**Overall Recommendation:** 4
**Confidence:** 2

**Summary:**

This paper proposes ReTreever, a new hierarchal retrieval model that augments a frozen encoder with a differentiable binary tree. Each node in the tree contains a trainable routing function, allowing both queries and documents to be organized into hierarchical paths from the root to the leaves. The tree is trained end-to-end using a contrastive loss, which encourages positive query-document pairs to follow similar paths while routing negatives to different branches. During inference, ReTreever provides two search strategies: (1) single-index search, which uses the node assignment probabilities at various depths as coarse-to-fine representations for vector-based similarity search; (2) multi-index search, which uses the learned leaf clusters to limit search to the top-k relevant partitions, similar to IVF in Faiss. Experiments across text, image, and audio datasets demonstrate ReTreever's effectiveness in balancing retrieval accuracy, latency, and interpretability.

**Compliance With Llm Reviewing Policy:**

Affirmed.

**Final Justification:**

The authors' rebuttal have addressed most of my concerns and I raise my rating

**Key Questions For Authors:**

1. What is the total training budget (time and computational cost) for ReTreever? More importantly, how many trainable parameters does the tree introduce compared to the frozen encoder? How does its parameter count and performance compare to other PEFT methods when applied to the same frozen encoder? Including a plot of retrieval performance vs. number of trainable parameters would greatly strengthen the paper's contribution.
2. Following the contrastive training, what does the distribution of documents across leaf nodes look like? Is the tree approximately balanced, or does the model exhibit collapse where a small number of leaves capture the vast majority of documents?
3. The paper uses the term "hierarchical retrieval," but the final retrieval step in both proposed methods is still based on flat vector similarity. Could the authors clarify what they consider to be the "hierarchical" aspect of the actual retrieval process?

**Limitations:**

yes

**Strengths And Weaknesses:**

Strengths:
1. Balancing latency and performance is crucial for practical retrieval methods in real-world information retrieval or recommender system applications. The idea of incorporating a learnable tree structure to model the inherent coarse-to-fine hierarchical organization of queries and documents is well-motivated. This offers valuable insights for future research in the field.
2. The experiments span a wide range of modalities (text, image, audio) and datasets, which validates the generalizability of ReTreever.

Weaknesses:
1. ReTreever does not alter the underlying vector search process. In the single-index setup, the final retrieval step still relies on standard vector similarity search over the derived probability vectors. Similarly, the multi-index search is functionally analogous to IVF in Faiss, where the tree merely provides a clustering mechanism. The core novelty lies in how the tree represents the data after fine-tuning, rather than in introducing a fundamentally new form of retrieval. As such, the paper is more accurately characterized as proposing a parameter-efficient fine-tuning (PEFT) method for a pre-trained bi-encoder, where the learnable tree injects a structured inductive bias. Consequently, the current baseline comparisons are insufficient. A broader range of fine-tuning approaches (e.g., PEFT methods like LoRA) should be considered to isolate the contribution of the tree structure itself. Including a comparison with a fully fine-tuned encoder would also help establish a stronger performance ceiling.
2. The comparison in Figure 4 is fundamentally unfair. It pits ReTreever's multi-index search (which uses a tree learned with a contrastive objective) against standard approximate search indexes (ScaNN, IVF) built on top of a frozen bi-encoder. This confounds two variables: the quality of the representation and the efficiency of the search structure. A fairer comparison would involve building the same approximate indexes (ScaNN, IVF) on top of ReTreever's own leaf-assignment representations and comparing that to ReTreever's multi-index search. This would isolate the contribution of the tree-based routing to search efficiency from the improved quality of the representation.
3. The paper does not provide crucial analysis regarding the learned tree structure itself. For instance, after contrastive training, is the tree balanced? Do all leaf nodes receive a roughly equal distribution of documents, or does the model collapse by funneling most data through a few paths? An analysis of the entropy of the routing probabilities or the distribution of documents per leaf would be beneficial to understand whether the tree is being utilized effectively.

---

> ### Author Rebuttal · Authors · 2026-03-31
>
> We thank the reviewer for highlighting the importance of latency–quality trade-offs, the value of a learnable coarse-to-fine tree, and the breadth of our text/image/audio evaluation.
>
> **W1** : We think the reviewer’s characterization of ReTreever as “PEFT on a frozen bi-encoder” captures only part of the picture. The main contribution is a retrieval-optimized hierarchy that simultaneously defines (i) coarse-to-fine representations, (ii) an inspectable semantic organization of the corpus, and (iii) the routing structure used for multi-index retrieval. Thus, the learned tree is not only an efficient adapter on top of the encoder, but also an extension that unlocks inspection and search. Likewise, although ReTreever multi-index and IVF both restrict search to selected partitions, there is a key difference: IVF applies task-agnostic clustering to a fixed embedding space, whereas ReTreever learns the hierarchy end-to-end for retrieval, and that same hierarchy also defines the embedding space used at inference. Under this framing, relevant baselines are other lightweight additions on top of the same frozen encoder, because these comparisons isolate whether gains come merely from added parameters or from the learned hierarchy itself. This is why we compare against Linear, MLP, and MRL on the same frozen backbone. A (fully or PEFT) fine-tuned encoder would be a useful raw-performance ceiling, but not directly comparable to any method in our frozen encoder regime.
>
> **W2**: The point of Fig. 4 is to assess the data accessed vs retrieval performance trade-off of ReTreever as compared to existing methods. It already goes beyond frozen-encoder baselines by including Encoder+MRL+IVF/ScaNN, where MRL is a learned coarse-to-fine adapter and thus allowing us to isolate the contribution of the learned hierarchy. We thank the reviewer for suggesting this useful complementary search setting for ReTreever: applying a standard ANN directly to ReTreever's representations to evaluate them in a flat search space. We therefore ran this control on ReTreever’s finest leaf-assignment vectors, and Fig. 5 [0] confirms that these learned vectors are strong embeddings under a conventional ANN backend. We view this as complementary to multi-index retrieval, since the two settings probe different questions: ANN on ReTreever vectors evaluates the learned representations in a flat search space, whereas multi-index evaluates both the learned representations and the learned tree as a router that restricts search to semantically similar leaves. This distinction matters at scale: on the 124.5M-context Reddit Title-Body corpus (Reviewer ttPY, W3/Q1, Fig. 4 [0]), ReTreever multi-index remained practical, whereas ScaNN exceeded 1 TB RAM and OOMed. We will revise the paper to clarify this distinction in the discussion of Fig. 4.
>
> **W3/Q2**: We agree this analysis is important, and our measurements indicate skew without collapse. ReTreever is not forced to allocate documents uniformly across leaves, but it also does not funnel the corpus into a few active paths. On NQ we observe 0 empty leaves and occupancies from 1 to 349 documents per leaf (mean 72, median 64); Fig. 2 [0] shows similarly broad utilization across datasets. Mean per-document leaf-routing entropy is 7.68–8.97 out of 10 max bits for 1024 leaves, again indicating substantial spread rather than hard collapse. We will add this analysis in the revision.
>
> **Q1**: ReTreever adds 3.7M trainable parameters on top of a frozen 66M DistilBERT-MS MARCO encoder. The main setup uses 200k steps, batch size 64, and H100 80GB GPUs. Median training times are 8.89 / 7.88 / 3.44 / 6.00h on 1×H100 for NQ / HotpotQA / RepLiQA / TopiOCQA, and 3.65 / 13.10h on 4 GPUs for ImageNet / VoxCeleb2, with the longer VoxCeleb2 runtime largely due to on-the-fly audio feature computation. For the same frozen encoder, the trainable parameter counts are Linear 0.6M, MRL 0.6M, MLP 4.7M, and ReTreever 3.7M. On NQ, the corresponding nDCG@10 values are 47.62, 46.60, 52.23, and 54.58. We will include this compute/parameter comparison in the revision.
>
> **Q3**: We use the term hierarchical retrieval because the hierarchy shapes both the representation and the search process. Even the finest ReTreever vectors are obtained by routing the input through all tree levels and combining node decisions via tree propagation, so each leaf-assignment vector depends on the full ancestor structure (Appendix I). The same tree also provides explicit coarse-to-fine embeddings at intermediate levels. In the multi-index setting, the hierarchy further serves as the retrieval structure by partitioning the corpus into leaf indices and routing each query to its most relevant leaves before similarity search. Thus, flat vector comparison is only the final scoring step; the pipeline itself is hierarchical end-to-end.
>
> [0] https://anonymous.4open.science/r/b-EDFB/e.pdf

---

> > ### Author Rebuttal · Reviewer_D6Se · 2026-04-04
> >
> > The authors' rebuttal have addressed most of my concerns and I raise my rating

---

> > > ### Author Response · Authors · 2026-04-05
> > >
> > > Thank you for your thoughtful reconsideration and for updating your score. We appreciate that our rebuttal addressed your concerns, and we are grateful for the time and care you gave to re-evaluating the paper.

---

### Official Review · Reviewer_ttPY · 2026-03-11

**Soundness:** 3
**Presentation:** 4
**Significance:** 3
**Originality:** 3
**Overall Recommendation:** 4
**Confidence:** 4

**Summary:**

This paper introduces ReTreever, a novel retrieval method based on the construction of a learned binary tree. This approach enables the generation of intermediate context representations (encompassing text, audio, and images) across multiple tree levels through a coarse-to-fine representation strategy. Within this hierarchy, descending the tree yields increasingly granular descriptions of the context. The representation is structured such that each vector dimension corresponds to the stochastic probability of node membership at a given level.

ReTreever offers the flexibility to balance computational cost and performance, providing an effective compromise between the high performance (but low efficiency) of hierarchical retrieval and the high efficiency (but lower performance) of flat retrieval approaches. The method is specifically designed to optimize the retrieval objective, ensuring that similar queries and contexts are mapped to the same nodes and leaves, while dissimilar ones are separated. This is achieved by optimizing a Softmax-based InfoNCE loss.

Furthermore, this tree construction implicitly facilitates semantic clustering, where the level of detail becomes increasingly specialized as the tree depth increases (e.g., broad categories such as 'Sports' or 'Science' at the upper levels refine into specific topics like 'Swimming' or 'Biochemistry' at deeper levels).

Results demonstrate that ReTreever achieves a superior trade-off between performance and latency among hierarchical methods

**Compliance With Llm Reviewing Policy:**

Affirmed.

**Final Justification:**

The authors have answered my questions during the rebuttal period. I do still believe the submission would benefit from a broader set of empirical tests (e.g., varying dimensionality), and as such am hesitant to raise the score to Accept.

**Key Questions For Authors:**

1. To what extent might the choice of different encoder architectures influence the observed performance in text retrieval? Specifically, would the advantages of ReTreever persist, or potentially disappear, if more modern or retrieval-optimized backbones were employed?

2. Do the current performance remain robust when scaling from millions to billions of documents?

3. Finally, could the authors provide insights into how the method performs across a broader range of tasks beyond standard Question-Answering (QA) datasets to demonstrate its generalizability?

**Limitations:**

Yes

**Strengths And Weaknesses:**

There are, in my opinion, one major category of weaknesses and one minor one. The former involves the empirical evaluation setup:

- The experiments included in the submission are centered around a representation size of 1024 and a 512-token input truncation. It's unclear to me whether these results hold across different dimensionalities. It'd be informative to conduct a systematic investigation of how performance scales or fluctuates with varying vector sizes.

- Another limitation is the scale at which the method is evaluated: While the authors position the method as an "at-scale" retrieval solution—a claim that is indeed supported by the implementation details—the datasets used in the work are limited to the million-document range. To fully validate the scalability of ReTreever, it would be highly beneficial to see how it performs against billions of documents (e.g., using the Microsoft Turing-ANNS benchmark) and how it differentiates itself from established baselines like ScaNN in such high-volume contexts.

- I should note that the text-retrieval experiments rely on DistilBERT, which, while widely used, is somewhat dated compared to the current state-of-the-art. It would be valuable to evaluate the framework using more recent encoders specifically optimized for retrieval tasks, such as Contriever, to see if the observed gains persist across different backbone architectures.


As for the relatively minor weakness: the authors need to put more effort into inserting citations and referencing existing material when they can. For example:
- The authors mention frameworks such as ScaNN and IVF but omit formal citations.
- The paper refers to Hier-KMeans and Hier-GMM without providing adequate explanations or references regarding their specific implementation. Adding context or citations for these methods is essential for the reader to fully grasp the comparative baseline analysis.
- A study that warrants mention is the work by Vecchiato et al., "A Learning-to-Rank Formulation of Clustering-Based Approximate Nearest Neighbor Search." Similar to the proposed method, their approach learns a routing function for clustering-based ANNS by training on query-context pairs to structure the search space.

---

> ### Author Rebuttal · Authors · 2026-03-31
>
> We thank the reviewer for their careful reading and for recognizing the novelty of ReTreever, its coarse-to-fine multimodal representations, its implicit semantic clustering, and its superior performance-latency trade-off among hierarchical retrieval methods.
>
> **W1**:  Although the main tables focus on the finest 1024-dimensional setting, the submission already includes three studies of robustness to representation size:
> (1) Figure 7 evaluates the full 1–1024-dimensional coarse-to-fine range across six datasets
> (2) Table 5 varies finest representation size through tree depth
> (3) Table 6 varies the number of level embeddings per level.
> Together, these results suggest that the gains are not tied to a single dimensionality or tree capacity. In addition, for the rebuttal, we ran a fourth study addressing the 512-token truncation concern: on a long-context LOCO/StackOverflow[1] subset with BGE-M3 and context length 8192, ReTreever improves over the base encoder from 19.74 to 26.14 nDCG@10 and from 45.16 to 53.96 Hit@10
>
> **W2/Q2**:  We thank the reviewer for noting that the paper’s implementation details already support our scalability claim, To address validation beyond the million-document regime, we ran an additional experiment on the Reddit Title-Body dataset [2], indexing the full **124.5M**-context corpus. At this scale, ScaNN on the encoder representations exceeded 1 TB of RAM during construction and failed with an OOM error, while ReTreever’s multi-index built and ran successfully under the same memory budget. Excluding representation preprocessing/loading, building the 1024 leaf-level indices took 14 minutes with 1024-dimensional indexed representations, and only 4 minutes with coarser 512 and 2 minutes with 256-dimensional tree representations. ReTreever learns a natural hierarchical partition, so retrieval reduces to exact L1 search within visited leaf partitions, while coarser tree representations further lower cost. Together with the data-access/quality tradeoff already shown in Fig. 4(b) [0], this highlights a practical scalability advantage over standard flat indexes.
>
> **W3/Q1**: We appreciate the reviewer’s suggestion to examine ReTreever with stronger retrieval-tuned backbones. The submission already includes a study with BGE (Appendix E), where ReTreever improves over the base encoder on NQ (60.46 vs. 55.57 nDCG@10) and TopiOCQA (27.82 vs. 22.19), and remains close on HotpotQA (91.05 vs. 91.54) and RepLiQA (85.74 vs. 86.42). We also ran Contriever experiments during rebuttal, where Table 1 [0] shows the same pattern: gains on 2/4 datasets (NQ, TopiOCQA) and competitive results on the other two. These results suggest that ReTreever is not tied to a single backbone, but acts as a lightweight **3.7M**-parameter hierarchical system that adds an inspectable semantic tree, coarse-to-fine representations, and controllable latency–quality trade-offs through scalable multi-index retrieval.
>
> **W4/W5/W6**: We thank the reviewer for pointing this out. In the revision, we will add formal citations for ScaNN [3], IVF[7] and clarify that Hier-KMeans and Hier-GMM are not prior named retrieval methods but our own top-down recursive baselines, obtained by repeatedly partitioning each node with k-means [4] or a Gaussian mixture model [5], respectively. We will also discuss the recent work of Vecchiato et al. [6], which learns routing functions for clustering-based ANN search from query-document supervision; unlike that line of work, ReTreever learns an end-to-end hierarchical representation that simultaneously supports coarse-to-fine embeddings, semantic inspectability, and multi-index retrieval.
>
> **Q3**: We thank the reviewer for raising this point. The evaluation already goes beyond QA: alongside four text QA benchmarks, we study image retrieval on ImageNet-1K, speech retrieval on VoxCeleb2, and, in the rebuttal, a non-QA text matching task on Reddit Title-Body (Fig 4. [0]). Together, these results show applicability beyond QA across modalities, and we will clarify this scope in the paper.
>
> [0] https://anonymous.4open.science/r/b-EDFB/e.pdf
>
> [1] Kanakarajan et al., "LOCO: The 88-billion Token Open Text Documents Benchmark" (2024)
>
> [2] Jason Baumgartner et al., "The Pushshift Reddit Dataset." ICWSM 2020
>
> [3] Guo, R et al. “Accelerating Large-Scale Inference with Anisotropic Vector Quantization.” ICML 2020
>
> [4] MacQueen, J. “Some Methods for Classification and Analysis of Multivariate Observations.” Proceedings of the Fifth Berkeley Symposium on Mathematical Statistics and Probability, Vol. 1, pp. 281–297, 1967
>
> [5] McLachlan, G. and Peel, D.,  Finite Mixture Models. Wiley, 2000
>
> [6] Vecchiato, T. et al,  “A Learning-to-Rank Formulation of Clustering-Based Approximate Nearest Neighbor Search.” SIGIR 2024, pp. 2261–2265
>
> [7]  Sivic and Zisserman. “Video Google: A Text Retrieval Approach to Object Matching in Videos.” ICCV 2003

---

> > ### Author Rebuttal · Reviewer_ttPY · 2026-04-02
> >
> > Thank you for your detailed note and additional discussion!

---

> > > ### Author Response · Authors · 2026-04-05
> > >
> > > Thank you very much for your acknowledgement and for noting that we addressed all of your concerns.
> > >
> > > We would be sincerely grateful if you would consider updating your score to reflect your revised assessment.

---

### Official Review · Reviewer_gMSz · 2026-03-12

**Soundness:** 3
**Presentation:** 4
**Significance:** 3
**Originality:** 3
**Overall Recommendation:** 5
**Confidence:** 3

**Summary:**

This paper proposes RETREEVER, a tree-based hierarchical retrieval method designed to balance retrieval efficiency, accuracy, and the interpretability (transparency) of the system's internal organization. Unlike recent high-cost hierarchical retrievers that rely on Large Language Models (LLMs) for construction and navigation (e.g., RAPTOR), RETREEVER operates entirely in the feature embedding space without requiring LLM calls. The method employs a cross-attention mechanism as the tree's split function and optimizes the structure end-to-end via an InfoNCE contrastive loss based on the negative Total Variation Distance (nTVD). Furthermore, the authors introduce a "stochastic depth training" strategy, enabling the model to learn coarse-to-fine multi-scale representations, which allows for a flexible trade-off between computational cost and retrieval accuracy during the inference stage.

**Compliance With Llm Reviewing Policy:**

Affirmed.

**Final Justification:**

The inclusion of the HNSW baseline, along with the clear reporting of wall-clock training time (~8.89 hours on a single H100) and the relatively small parameter overhead (~3.7M), substantially strengthens the paper’s practical relevance. The authors are encouraged to incorporate all these improvements into the final version.

**Key Questions For Authors:**

1. Routing Dynamics: During the early stages of tree optimization, does the contrastive loss cause routing probabilities to "harden" too quickly (leading to idle or dead nodes)? Did you employ specific techniques like temperature scaling or load-balancing regularization to ensure sufficient exploration of the tree structure?

 2. Topology Flexibility: The current framework utilizes a perfectly balanced binary tree with a fixed depth. Is this rigid topology suboptimal for datasets with severe long-tailed distributions? Are there plans to support data-driven or unbalanced tree topologies in future work?

 3. Dynamic Updates: After the tree parameters $\theta$ are frozen, how does RETREEVER handle the insertion of new, out-of-distribution (OOD) documents? If semantic drift occurs within the corpus, is a global retraining of the tree strictly required?

 4. HNSW Baseline: Why was HNSW excluded from the multi-index approximate search experiments (Figures 4 and 8)? Under extremely low latency requirements, how does RETREEVER compare to a highly optimized HNSW implementation in terms of the data access rate vs. recall trade-off?

 5. Wall-clock Training Time and Computational Budget: While the appendix mentions the use of a single NVIDIA H100 80GB GPU for 200,000 steps with a batch size of 64, the actual wall-clock training time (or total GPU hours) is not reported. Given the efficiency claims compared to LLM-dependent methods, providing the exact end-to-end training time would help clarify the practical scalability and computational budget of RETREEVER. Could the authors provide these concrete time metrics?

 6. Granular Details of Parameter Sharing Mechanisms: The paper discusses three distinct split functions (Linear, MLP, and Cross-Attention). I would appreciate more granular details on exactly how parameter sharing is implemented across them. Specifically for the Cross-Attention formulation, could the authors clarify the exact dimensionalities and the forward pass flow involving the globally shared projection matrices ($W_q, W_k, W_v$), the level-specific embeddings ($e_l$), and the node-specific linear mappings? A clear breakdown of the parameter counts and the detailed sharing topology for each routing mechanism would greatly enhance the reproducibility and clarity of the architectural design.

**Limitations:**

No. While the authors mention future work on incremental updates, the current discussion on limitations could be expanded. Specifically, it would be helpful to discuss the potential performance bottlenecks of a fixed, perfectly balanced tree topology when applied to highly skewed datasets. Additionally, acknowledging the true computational training overhead (in terms of wall-clock time) and the memory boundaries introduced by the cross-attention sharing mechanisms would provide a more complete and transparent picture of the method's practical constraints.

**Strengths And Weaknesses:**

* **Strengths:**
  * The paper proposes a novel and interesting differentiable approach to hierarchical retrieval by utilizing cross-attention mechanisms for routing.

  * The empirical evaluations demonstrate impressive latency speedups (e.g., ~160x faster than RAPTOR) and competitive accuracy across diverse modalities (text, image, audio).


* **Weaknesses:**
  * **Methodological Rigidity (Topology):** A primary concern lies in the assumption of a hardcoded, perfectly balanced binary tree (e.g., depth 10). While computationally convenient, this rigid topology may struggle with real-world, long-tailed data distributions, potentially leading to capacity mismatches. The paper would benefit from discussing how the framework could adapt to unbalanced or data-driven topologies.

  * **Missing Computational Context:** While the paper emphasizes efficiency over LLM-dependent methods, it lacks reporting on the actual wall-clock training time or total GPU hours required to optimize the entire tree. Providing this end-to-end computational budget is necessary to fully contextualize the scalability claims.

  * **Architectural Clarity (Parameter Sharing):** The exact implementation details of parameter sharing—especially for the Cross-Attention split function—could be significantly clarified. A clearer breakdown of how the globally shared projection matrices, level-specific embeddings, and node-specific mappings interact, along with their respective parameter counts, would greatly improve reproducibility.

  * **Baseline Selection:** The evaluation in the approximate multi-index search experiments (Figures 4 and 8) would be more convincing if it included comparisons against other AkNN methods (e.g., HNSW, PQ, RaBitQ, Vamana).

  * **Terminology Precision:** The repeated use of the term "Transparency" feels slightly overstated. Given that the cross-attention routing mechanism remains a neural black box, describing the contribution as providing post-hoc structural interpretability (via semantic clusters) would be more precise.

---

> ### Author Rebuttal · Authors · 2026-03-31
>
> We thank the reviewer for the constructive feedback and for recognizing the novelty of the differentiable hierarchical routing and the strong latency/accuracy trade-off, impressive speedups over prior hierarchical methods, and competitive accuracy over diverse modalities.
>
> **W1/Q1/Q2**: We agree that fixed balanced topology should be discussed more explicitly. A balanced tree does not assume uniform document allocation: routing is free to use leaves non-uniformly. The data are already skewed before tree learning: in a 1024-way k-means partition of frozen encoder embeddings, the largest clusters contain 774/608/334/214 documents for NQ/HotpotQA/RepLiQA/TopiOCQA, versus means of 72/89/69/30 (Fig. 1 [0]). The learned tree is also non-uniform but does not collapse: on NQ the largest leaf has 349 documents and there are 0 empty leaves (Fig. 2 [0]). We do not use an explicit routing-temperature schedule or load-balancing regularizer; collapse is instead discouraged by the bidirectional InfoNCE objective with many in-batch negatives and a learnable softmax temperature. We will add node/leaf utilization statistics in the appendix. We agree that learned unbalanced or adaptive topologies are natural extensions; e.g., one could split overloaded leaves or learn topology jointly as in prior differentiable-tree work [1,2].
>
> **W2/Q5**:  We agree that wall-clock cost should be reported explicitly. Using one H100 80GB GPU, median training times were 8.89h / 7.88h / 3.44h / 6.00h on NQ / HotpotQA / RepLiQA / TopiOCQA. On ImageNet-1K and VoxCeleb2 they were 3.65h and 13.10h using 4 GPUs; VoxCeleb2 is slower mainly because audio features are computed on the fly.
>
> **W3/Q6**: We will clarify the parameter-sharing scheme in Appendix 1. Briefly, at each tree level, learnable level embeddings act as queries over frozen encoder token representations via a multi-head cross-attention split function; their contextualized outputs are mean-pooled into a shared level representation, and each node at that level applies its own learned linear map to produce a node-specific routing score. The projection matrices Wq, Wk, and Wv are shared across the entire tree. In total, Retreever adds 3.7M trainable parameters for the configuration reported in Table 11.
>
> **W4/Q4**:  We agree that the ANN comparison should be broader, and we therefore added HNSW as an additional graph-based ANN baseline. While HNSW improves retrieval quality at very low data-access budgets, in the more realistic 10–20% range it is similar to IVF, and RETREEVER-multi-index's results remain stronger here (Fig. 3 [0]). HNSW is also complementary to ReTreever’s multi-index setup: replacing the flat L1 search inside the selected leaves with per-leaf HNSW yields a stronger two-stage retriever, reaching ~ 54.0 / 84.5 / 80.0 / 30.8 NDCG@10 on NQ / HotpotQA / RepliQA / TopiOCQA at ~ 10% access vs. ~ 43.0/72.0/73.0/19.0 for the strongest encoder-based HNSW baseline  (Fig. 3 [0]). PQ and RaBitQ are quantization-based methods and would require a separate compression-focused comparison, which we will note in the revision.
>
> **W5**: We appreciate this suggestion and agree that structural interpretability is the more precise term. Our claim is not mechanistic transparency of the neural router, but that ReTreever is trained to produce an inspectable hierarchical organization of the corpus; this usage is also consistent with prior ML literature on structural interpretability [3]
>
> **Q3**:  For new in-distribution documents, ReTreever does not require retraining: documents are encoded with the frozen encoder, routed through the learned tree, and inserted into the corresponding indices. The harder case is when enough new documents arrive that the corpus semantics themselves shift and the learned hierarchy should evolve. In that regime, the current tree remains usable, but it may no longer be the optimal semantic organization, since it was learned for the original document distribution. This is a broader dense-retrieval issue rather than something unique to ReTreever; e.g., BEIR notes that strong dense retrievers can generalize poorly under distribution shift [4]. We could think of extensions to ReTreever to efficiently adapt it to data shifts, such as local tree updates, lightweight distribution-specific adapters, or topology modifications that selectively refine overloaded or shifted regions of the tree. These are all promising directions for future work, and we will clarify our current scope in the revision.
>
> We will expand the limitations section to reflect the discussion here more explicitly.
>
> [0] https://anonymous.4open.science/r/b-EDFB/e.pdf
>
> [1] Zantedeschi et al., ICML 2021, arXiv:2010.04627
>
> [2] Tanno et al., ICML 2019, arXiv:1807.06699
>
> [3] Lin et al., ICML 2021, arXiv:2012.11589
>
> [4] Thakur et al., 2021, arXiv:2104.08663

---

> > ### Author Rebuttal · Reviewer_gMSz · 2026-04-02
> >
> > I thank the authors for their comprehensive and convincing rebuttal, which perfectly addressed my initial concerns. The addition of the HNSW baseline, along with the transparent disclosure of wall-clock training times (~8.89h on a single H100) and the minimal parameter overhead (~3.7M), significantly strengthens the paper's practical value. Given these robust updates and the authors' scholarly rigor, I am pleased to raise my score.

---

> > > ### Author Response · Authors · 2026-04-02
> > >
> > > Thank you for the very positive follow-up. We are grateful that the rebuttal resolved your concerns and that you indicated you would raise your score.
> > >
> > > We noticed that the score currently displayed in the system has not changed, so we wanted to politely mention this in case the update was missed.

---

### Decision · Program_Chairs · 2026-04-30

**Decision:**

Accept (regular)

**Comment:**

The paper proposes a hierarchical retrieval approach that learns a tree structure over frozen encoders. It has received four reviews: one Accept, two Weak Accepts, and one Weak Reject. Overall, reviewers are positive about the paper, and most concerns have been addressed in the rebuttal.

The reviewers highlight several strengths: (i) well-motivated, addressing relevant problem (ii) the hierarchical design is intuitive and supports effective coarse-to-fine representations (iii) strong  empirical evaluation spanning multiple datasets/situations and (iv) methodology demonstrates a clear and practical utility  while also offering a degree of structural interpretability.

While some concerns remain, mainly as suggestions for improvements in evaluation, clarification of certain experimental comparisons and design choices,  the overall consensus is clearly positive, and the paper is recommended for acceptance.